# Using domain knowledge for robust and generalizable deep learning-based CT-free PET attenuation and scatter correction

Rui Guo[1,2,8], Song Xue[3,8], Jiaxi Hu[3], Hasan Sari[3,4], Clemens Mingels[3], Konstantinos Zeimpekis[3], George Prenosil[3], Yue Wang[1,2], Yu Zhang[1,2], Marco Viscione[3], Raphael Sznitman[5,6], Axel Rominger[3], Biao Li[1,2] ✉ & Kuangyu Shi[3,6,7]

Despite the potential of deep learning (DL)-based methods in substituting CT-based PET attenuation and scatter correction for CT-free PET imaging, a critical bottleneck is their limited capability in handling large heterogeneity of tracers and scanners of PET imaging. This study employs a simple way to integrate domain knowledge in DL for CT-free PET imaging. In contrast to conventional direct DL methods, we simplify the complex problem by a domain decomposition so that the learning of anatomy-dependent attenuation correction can be achieved robustly in a low-frequency domain while the original anatomy-independent high-frequency texture can be preserved during the processing. Even with the training from one tracer on one scanner, the effectiveness and robustness of our proposed approach are confirmed in tests of various external imaging tracers on different scanners. The robust, generalizable, and transparent DL development may enhance the potential of clinical translation.

Ionizing radiation burden is a major concern in the practice of positron emission tomography/computed tomography (PET/CT) imaging, which hampers its application in many situations[1,2]. Although the current practice of PET/CT imaging according to the guidelines[3,4] routinely uses low-dose CT, a considerable contribution originates from low-dose CT (6.4 mSv[5,6]), which is typically used for the correction of attenuation (AC) and anatomical localization in PET imaging. And the burden from CT becomes more evident with the advent of the long axial field of view (LAFOV) total-body PET scanners, which enable unprecedented levels of image quality and quantification accuracy with reduced radiopharmaceutical dose[7]. Although ultra-low dose attenuation correction CT (2.1 mSv, without anatomical localization)[5]

imaging significantly reduces the radiation exposure, eliminating AC CT, i.e., CT-free PET imaging, is beneficial in a number of situations such as pediatric examinations[1], multiple PET/CT examinations[8–10], pharmaceutical tests[11,12], and so on, where additional radiation burden due to redundant anatomic imaging could be concerned.

Therefore, numerous research efforts have been devoted to developing methods for CT-free PET correction. Magnetic resonance (MR)-based approaches yielded satisfactory results in brain PET[13]. Alternatively, attenuation maps (μ-maps) can also be derived using the maximum likelihood estimation of activity and attenuation (MLAA) algorithm. MLAA can be further improved with the use of the additional time of flight (TOF)[14,15]. Due to the insufficient timing resolution

[1]Department of Nuclear Medicine, Ruijin Hospital, Shanghai Jiao Tong University School of Medicine, Shanghai, China. [2]Collaborative Innovation Center for Molecular Imaging of Precision Medicine, Ruijin Center, Shanghai, China. [3]Department of Nuclear Medicine, Inselspital, Bern University Hospital, University of Bern, Bern, Switzerland. [4]Advanced Clinical Imaging Technology, Siemens Healthcare AG, Lausanne, Switzerland. [5]ARTORG Center, University of Bern, Bern, Switzerland. [6]Center of Artificial Intelligence in Medicine (CAIM), University of Bern, Bern, Switzerland. [7]Computer Aided Medical Procedures and Augmented Reality, Institute of Informatics I16, Technical University of Munich, Munich, Germany. [8]These authors contributed equally: Rui Guo, Song Xue. ✉e-mail: lb10363@rjh.com.cn

**Table 1 | Information on patients' demographics and diagnosis**

| Source | Training | Valid | Cross scanner | | | Cross tracer | | | |
|---|---|---|---|---|---|---|---|---|---|
| | SIEMENS Healthineers Vision 450 (SH)- 18F-FDG | SIEMENS Healthineers Vision 450 (SH)- 18F-FDG | SIEMENS Healthineers Vision 600 (Bern)- 18F-FDG | UI uMI 780 (SH)- 18F-FDG | GE Discovery MI (SH)- 18F-FDG | SIEMENS Healthineers Vision 450 (SH)- 68Ga-FAPI | SIEMENS Healthineers Vision 450 (SH)- 68Ga-DOTA-TATE | SIEMENS Healthineers Vision 600 (Bern)- 68Ga-DOTA-TOC | SIEMENS Healthineers Vision 600 (Bern)- 18F-PSMA |
| Number of patients | 470 | 51 | 62 | 98 | 104 | 7 | 17 | 8 | 12 |
| Total dose (MBq) | 337.8±63.4 | 305.1±66.0 | 254.4±51.7 | 265.8±32.1 | 336.0±66.7 | 152.9±21.2 | 95.0±17.3 | 152.0±7.3 | 241.6±15.5 |
| Post-injection time (min) | 75.7±32.1 | 82.3±18.6 | 78.4±13.4 | 66.4±20.1 | 80.9±27.5 | 54.1±7.3 | 60.1±18.7 | 96.9±20.1 | 152.7±23.3 |
| Gender (Male/Female) | 263/207 | 25/26 | 27/35 | 49/49 | 51/53 | 5/2 | 9/8 | 5/3 | 12/0 |
| Age (Year) | 58.2±14.3 | 56.8±19.0 | 64.0±14.1 | 59.4±13.5 | 57.6±13.0 | 59.0±14.7 | 41.4±14.0 | 60.9±15.0 | 74.5±6.4 |
| Weight (kg) | 62.8±10.9 | 62.2±13.2 | 71.5±13.7 | 61.3±12.0 | 61.3±11.1 | 63.1±11.6 | 69.6±15.5 | 80.9±23.4 | 84.9±14.2 |

of current clinical PET systems, the MLAA suffers from the crosstalk between the activity and attenuation distribution and high noise[16]. Recent MLAA methods alleviate the low-frequency crosstalk problem[17], but may over-smooth or over-estimate in the bone structure[18,19]. TOF-MLAA-based approaches using an external source like rod[20] or lutetium oxyorthosilicate background transmission[16,21] are under development to overcome the limitations.

Inspired by the rapid expansion of deep learning (DL)-based methods in various medical image analysis applications[22], many DL-based approaches for the CT-free PET imaging have been proposed, especially utilizing DL techniques[15,23,24]. One proposed approach was to generate μ-maps or pseudo-CT from non-corrected PET images[25], another kind is to directly generate corrected PET images from non-corrected ones[26]. However, a critical bottleneck of these DL methods is the limited capability in the application of heterogeneous domains of PET imaging. The spatial resolution and sensitivity vary between different scanners. More importantly, emerging tracers are evolving with different biodistributions. The rapid development in PET imaging makes it impossible to enumerate the heterogeneous application domain in the training data, which hampers the robustness and trustworthiness of DL-based AC methods.

To overcome the limitations of conventional DL-based techniques, we propose to employ a simple way to integrate domain knowledge in deep learning for CT-free PET imaging. We decomposed the complex end-to-end generation into two components, anatomy-independent textures (relating to tracers and diseases) and anatomy-dependent correction. Compared to direct approaches, estimation of only low-frequency anatomy-dependent correction using a 3D deep neural network can be more efficient and robust.

## Results
### Test on external scanners
The proposed DL algorithm was developed based on 470 subjects who underwent $^{18}$F-fluorodeoxyglucose ($^{18}$F-FDG) PET and scanned with Biograph Vision 450 (Vision 450, Siemens Healthineers) in Shanghai (SH). We tested the trained DL model with four test datasets to evaluate the robustness, which include 51 subjects from Vision 450, 98 subjects from UI uMI 780 (uMI 780, United Imaging), 104 subjects from GE Discovery MI (DMI, General Electric Healthcare) collected at SH, as well as 62 subjects from Biograph Vision 600 (Vision 600, Siemens Healthineers) collected at Bern. Detailed patient demographics are given in Table 1.

Figure 1a–c provides quantitative accuracy of the DL attenuation scatter corrected PET (DL ASC-PET) images to the original CT-based attenuation scatter corrected PET (CT ASC-PET) images on all four scanners. Our proposed domain knowledge integrated Decomposition-based DL was compared to two other direct 2D and 3D DL methods designed in a traditional end-to-end generation manner[27,28], which generate ASC-PET directly from non-attenuation and non-scatter corrected images (NASC-PET) with either 2D or 3D network. As shown in Fig. 1a–c, all three DL methods were capable of some degree of attenuation and scattering correction for different scanners, but Decomposition-based DL significantly outperformed the other two on all scanners ($p < 0.025$). Specifically, in terms of normalized root mean squared error (NRMSE), Decomposition-based DL improved 47.5% over Direct 2D and 49.1% over Direct 3D on Vision 450, and 60.0% over Direct 2D and 58.4% over Direct 3D on Vision 600, while on both scanners maintained a similar level of error ($p = 0.88$). When applied to DMI and uMI 780, Decomposition-based DL still outscored the other two by more than 20%. Results of peak signal-to-noise ratio (PSNR) and structural similarity index measurement (SSIM) of $^{18}$F-FDG imaging on the four different scanners showed the same tendency as the NRMSE results. Furthermore, we measured clinical imaging parameters such as $SUV_{mean}$, $SUV_{max}$, total lesion metabolism, as well as the most relevant radiomics features within the sphere

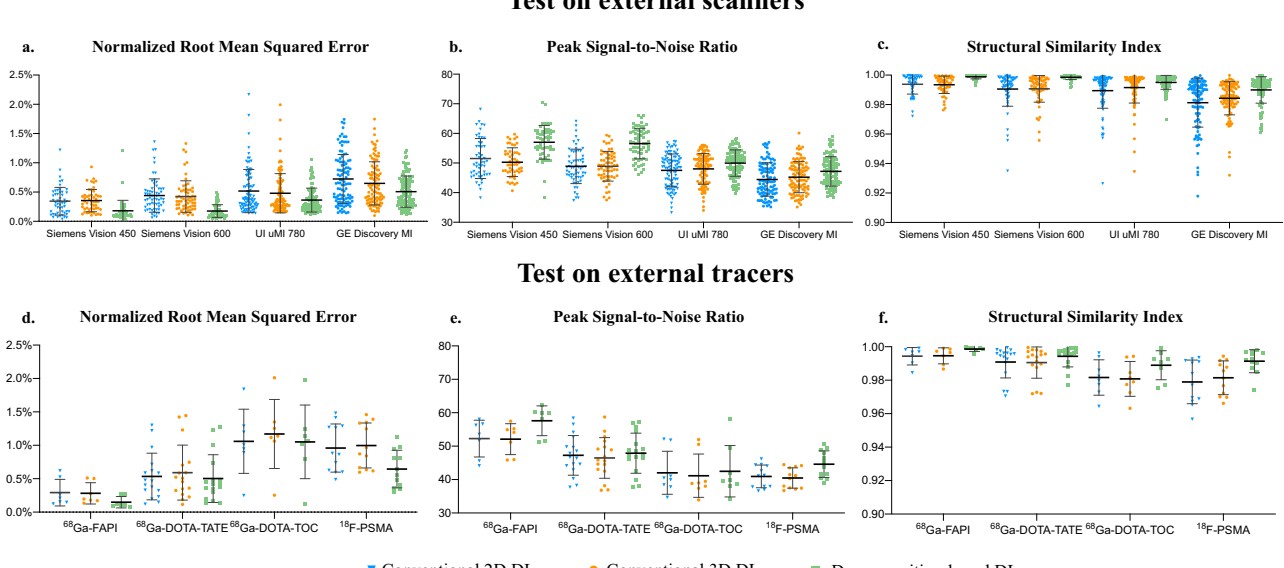

**Fig. 1 | Quantitative accuracy evaluated with global physical metrics on external scanners and external tracers.** Quantitative accuracy of the DL ASC-PET images generated with our proposed method, to the CT ASC-PET on the cross-scanner (**a**–**c**) and cross-tracer (**d**–**f**) settings, evaluated with global physical metrics including normalized root mean squared error (NRMSE), peak signal-to-noise ratio (PSNR) and structural similarity index measurement (SSIM). Data are presented as mean values +/− SD. Sample size: **a**–**c**: Siemens Vision 450 ($n = 51$), Siemens Vision 600 ($n = 62$), UI uMI 780 ($n = 98$), GE Discovery MI ($n = 104$); **d**–**f**: $^{68}$Ga-FAPI ($n = 7$), $^{68}$Ga-DOTA-TATE ($n = 17$), $^{68}$Ga-DOTA-TOC ($n = 8$), $^{18}$F-PSMA ($n = 12$).

volume of interest (VOI) of different organs (liver, kidney, and heart)[29–35]. Figure 2a−d shows the mean absolute percentage error (MAPE) of each feature on all scanners, calculated in reference to the CT ASC-PET, which demonstrates that Decomposition-based DL outperformed the other two on all scanners regarding local metrics as well. Activity distribution in the organs of interest confirmed the advantage of Decomposition-based DL, shown in Supplementary Fig. 2.

In addition to the quantitative evaluation, as shown in Fig. 3, we showed a head-to-head comparison of a representative imaging example of NASC-PET, DLASC-PET of all three methods, and CT ASC-PET, as well as joint histogram analysis depicting the correlation between activity concentration of DL ASC-PET and NASC-PET versus reference CT ASC-PET in Fig. 4. The Decomposition-based DL provided image quality comparable with CT ASC-PET and preserved more detailed information and less noise was observed compared to Direct 2D and 3D. The joint histogram analysis of an exemplary subject (Fig. 4a) exhibited voxel-wise similarity between reference CT ASC-PET and Decomposition-based DL ASC-PET with slopes of 0.94, 0.97, 1.05, and 0.94 for Vision 450, Vision 600, uMI 780 and DMI respectively. Voxel-wise absolute percentage error map of an exemplary subject depicting the difference between DL ASC-PET and reference CT ASC-PET are shown in Supplementary Fig. 3.

### Test on external tracers
The DL algorithm we developed, which is based on 470 subjects with $^{18}$F-FDG PET, was further applied to four external test datasets to evaluate the robustness on the cross-tracer settings, which included 7 subjects with $^{68}$Ga-fibroblast-activation protein inhibitors ($^{68}$Ga-FAPI) and 17 subjects with $^{68}$Ga-DOTA-Tyr-octreotate ($^{68}$Ga-DOTA-TATE) from Vision 450, as well as 8 subjects with $^{68}$Ga-DOTA-Tyr(3)-octreotide ($^{68}$Ga-DOTA-TOC) and 12 subjects with $^{18}$F-prostate-specific membrane antigen ($^{18}$F-PSMA) from Vision 600.

Figure 1d−f provides quantitative accuracy of the DL ASC-PET to the CT ASC-PET on four tracers. All three DL methods were capable of some degree of attenuation and scattering correction for different tracers, but Decomposition-based DL significantly outperformed the other two on $^{68}$Ga-FAPI and $^{18}$F-PSMA ($p < 0.025$). Specifically, in terms of normalized root mean squared error (NRMSE), Decomposition-based DL improved 49.0% over Direct 2D and 47.1% over Direct 3D on $^{68}$Ga-FAPI, and 32.3% over Direct 2D and 35.3% over Direct 3D on $^{18}$F-PSMA, while the advantage on $^{68}$Ga-DOTA-TATE and $^{68}$Ga-DOTA-TOC was less evident. Results of PSNR and SSIM showed the same tendency as the NRMSE results. Figure 2e−h shows the MAPE of each feature on all tracers, calculated in reference to the CT ASC-PET, which demonstrates that Decomposition-based DL outperformed the other two on all tracers regarding local metrics.

As shown in Fig. 3, the head-to-head comparison of a representative imaging example showed that the Decomposition-based DL provided image quality comparable with CT ASC-PET, preserved more detailed information and less noise was observed compared to Direct 2D and 3D. The joint histogram analysis of an exemplary subject (Fig. 4b) exhibited voxel-wise similarity between reference CT ASC-PET and Decomposition-based DL ASC-PET with slopes of 1.1, 1.0, 0.97, and 0.9 for $^{68}$Ga-FAPI, $^{68}$Ga-DOTA-TATE, $^{68}$Ga-DOTA-TOC and $^{18}$F-PSMA respectively. Voxel-wise absolute percentage error map of an exemplary subject depicting the difference between DL ASC-PET and reference CT ASC-PET are shown in Supplementary Fig. 3.

## Discussion
The advances in PET imaging technology have enabled unprecedented levels of image quality, even with reduced radiopharmaceutical dose[36].While the injected PET radiation dose can be reduced[7,37], the CT radiation dose accounts for an increasing proportion of the total PET/CT dose. However, accurate AC and SC are essential for image quality and precise PET quantification[38], clinically established CT-based approaches inevitably introduce ionizing radiation to patients[39]. This in turn renders a CT-less approach particularly attractive. In certain clinical and research scenarios, it is conceivable that PET examinations can be performed without the necessity of additional CT imaging, e.g., multiple PET tracers examinations[8–10], pediatric patients with previously acquired anatomic images[1], and pharmaceutical developments[11,12,40]. In these cases, radiation exposure could be reduced by omitting CT scans. For example, ionizing radiation

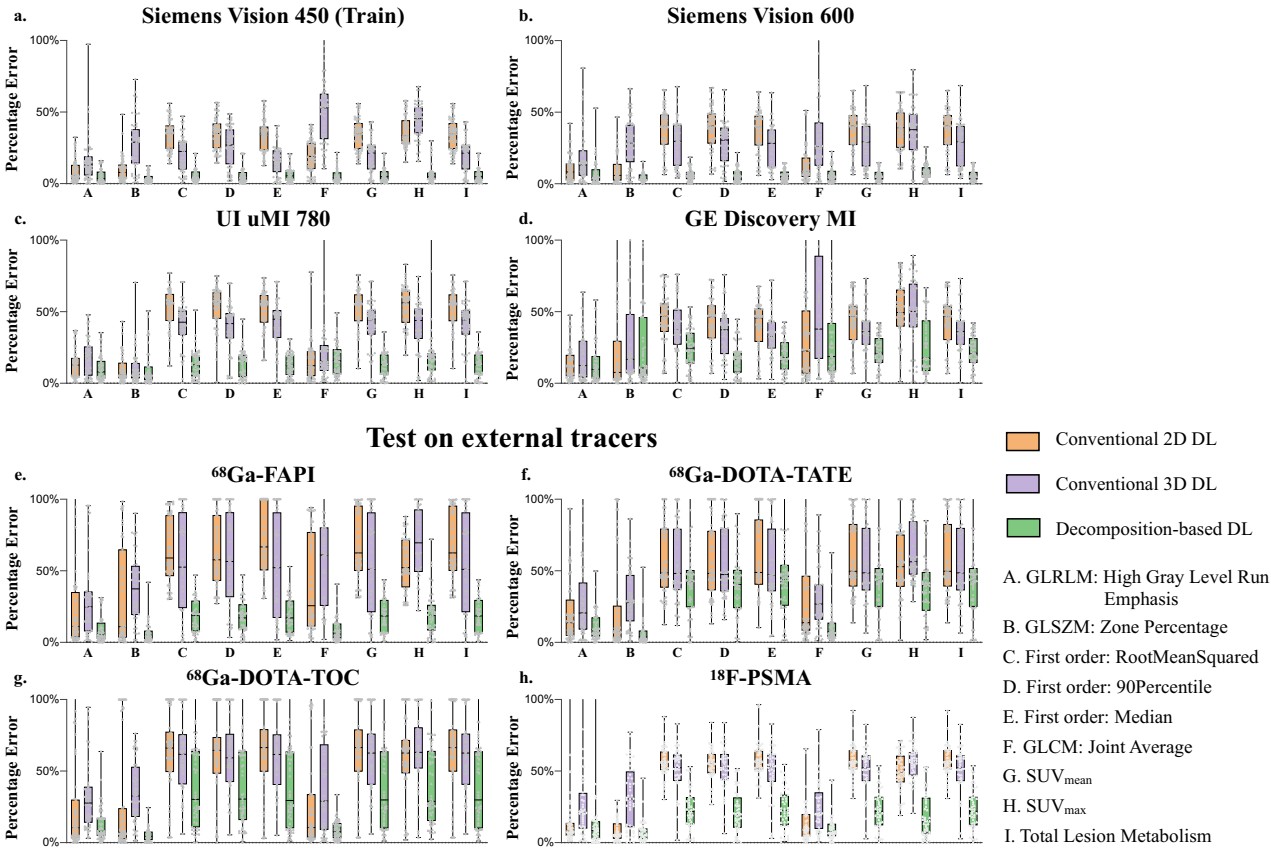

**Fig. 2 | Quantitative accuracy evaluated with local metrics on external scanners and external tracers.** Quantitative accuracy of the DL ASC-PET images generated with our proposed method, to the CT ASC-PET on the cross-scanner (**a**–**d**) and cross-tracer (**e**–**h**) setting, evaluated with clinical imaging parameters such as SUV$_{mean}$, SUV$_{max}$, total lesion glycolysis (TLG), as well as the most relevant radiomics features. Data are summarized by a box and whisker plot (the central line in the box plot indicates the median of the data, while the edges of the box indicate the 25th and 75th percentiles; extending from the box are whiskers, the top whisker expands to the maxima and the bottom whisker to the minima). Sample size: **a**–**d**: Siemens Vision 450 ($n = 51$), Siemens Vision 600 ($n = 62$), UI uMI 780 ($n = 98$), GE Discovery MI ($n = 104$); **e**–**h**: $^{68}$Ga-FAPI ($n = 7$), $^{68}$Ga-DOTA-TATE ($n = 17$), $^{68}$Ga-DOTA-TOC ($n = 8$), $^{18}$F-PSMA ($n = 12$).

exposure associated with multiple CT scans is of concern in pediatrics, where risk-benefit analysis doesn't favor CT scans and they have a considerable lifetime risk to develop secondary cancers[41]. Drug development is another example, PET potentially provides quantitative information about pharmacokinetics and pharmacodynamics as well as evidence if a drug is relevant within a disease population which is key to the proposed mode of action of a drug[11]. In the cases such as the development of $^{89}$Zr-labelled antibodies PET imaging, a series of PET/CT scans are usually performed within 2 weeks after the injection[42]. Repeated AC CT scans could be critical concerns in such kinds of investigations[40,43].

In contrast to anatomical imaging, PET imaging exhibits a large heterogeneity in its application domain. The development of instrumentation continues to improve the physical characteristics such as spatial resolution, sensitivity, and anatomical coverage which can enhance the texture details in the images[44], and the robustness of DL-based AC methods may be affected by changes in these physical characteristics. Furthermore, emerging tracers are introduced in PET imaging frequently[44] and are being evaluated in clinical trials. Different tracers exhibit different pharmacokinetics leading to a variety of bio-distributions. A proper DL-based AC method should be applicable for this large variety of scanners and tracers. The test results of the proposed method demonstrated that the Decomposition-based DL method is generally applicable for external scanners and tracers, which have not been touched during the training. The robust application in

different scanners and tracers illustrates the potential of clinical adoption of the proposed method. The credibility in the application of clinically established tracers can be improved with texture preservation. The extensibility of new tracers meets the demand for reduced radiation burden in the clinical tests of tracer developments.

With the increasing awareness of potential pitfalls of DL, reproducibility, and generalizability to previously unseen scenarios play a critical role in the credibility of any DL methodology, which mandates the robust and generalizable DL development[45]. Incorporating domain knowledge into the design of DL techniques is an alternative strategy to traditional techniques, studies have already achieved great success in disease diagnosis, lesion and abnormality detection, and segmentation[46,47]. The visualization of the data and statistical analysis demonstrates how our domain knowledge integrated DL method overcomes a well-known issue of conventional DL techniques. Without well-designed regularization and penalty system, the predictions could overfit the training data and may not be consistent with the governing physical laws.

Quantitative accuracy drop in PET describes the loss of detected photon pairs due to photon scattering and photoelectric absorption induced by the presence of dense material along lines of response (LOR), and the wrong LOR assigned following path change of scattered photons within the acceptance energy window requires to scatter correction. The main challenge of attenuation correction lies in finding reliable attenuation-correction factors (ACF) compensating for this

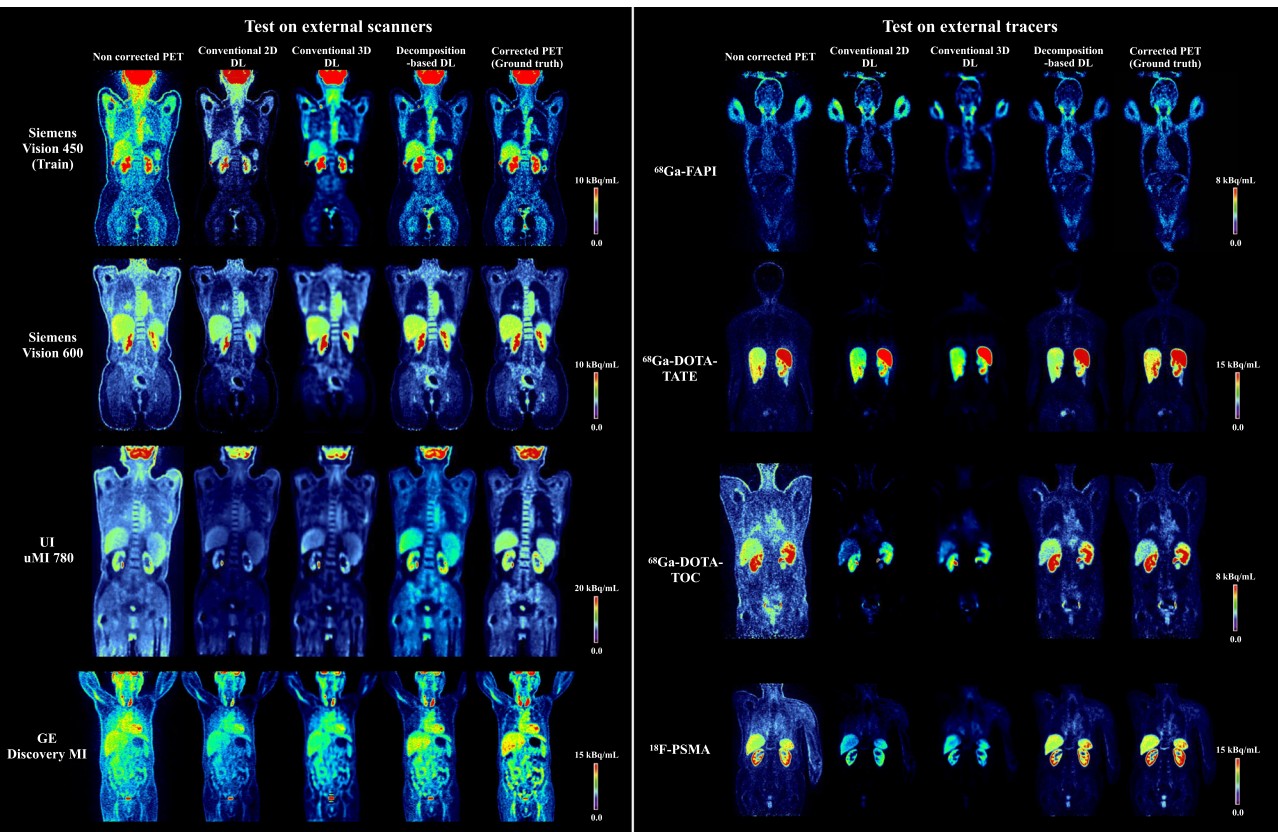

**Fig. 3 | Exemplary test results of external scanners and external tracers.** Exemplary test results of $^{18}$F-FDG imaging from Siemens Vision 450, Siemens Vision 600, UI uMI 780, and GE Discovery MI and imaging from $^{68}$Ga-FAPI (Vision 450-SH), $^{68}$Ga-DOTA-TATE (Vision 450-SH), $^{68}$Ga-DOTA-TOC (Vision 600-Bern) and $^{18}$F-PSMA (Vision 600-Bern). Note that the color bars used for non-corrected PET are in the range of 15% of the presented color bars.

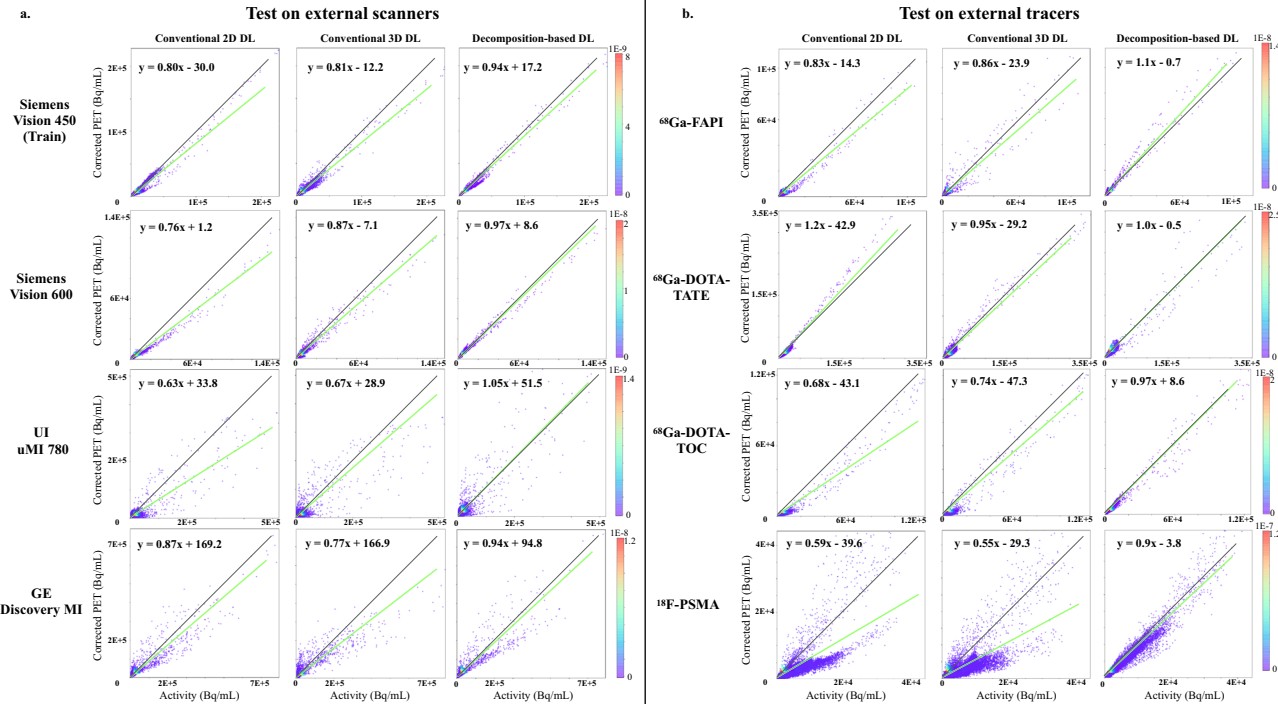

**Fig. 4 | Joint histogram analysis of exemplary test results.** Joint histogram analysis of an exemplary subject depicting the correlation between activity concentration of DL ASC-PET and NASC-PET versus reference CT ASC-PET for different scanners (**a**) and tracers (**b**).

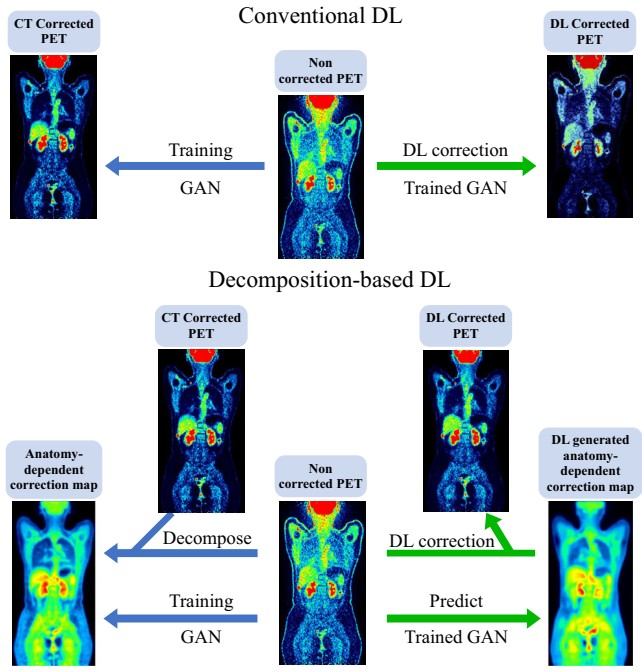

**Fig. 5 | General protocol of our proposed domain knowledge integrated deep learning method.** We decomposed the complex end-to-end generation from non-attenuation and non-scatter corrected images (NASC-PET) to corrected images (ASC-PET) into two components, anatomy-independent textures (relating to tracers and diseases) and anatomy-dependent correction, and regularized a 3D deep neural network to estimate this low-frequency anatomy-dependent correction information only.

loss, which is often calculated from a μ-map. The μ-map may also be used to compute an estimate of the scatter contribution within the unscattered PET emission data. Related studies reported DL-based approaches have been proposed for PET correction, such as the conversion of MR images to a pseudo-CT or μ-map[48–50], derivation of μ-map from NASC-PET[25]. These approaches still suffer from errors due to spatial mismatch between the emission and attenuation map data or require additional information or extra reconstruction procedures. Several emission-based approaches proposed the direct conversion of non-attenuation corrected PET images to the attenuation corrected images using convolutional encoder-decoder neural networks[23,24]. The applications of the emission-based approach are not limited to PET/ CT, but also include PET/MR as well as stand-alone PET systems (e.g., brain-dedicated PET). However, the direct approaches either suffer from the loss of spatial information (2D slice-based[27,28]), 3D patch-based[51,52], or the lack of voxel-wise detailed information (3D volume-based), which was verified in tests of the valid dataset (Vision 450-FDG). Figure 3 illustrated that the conventional 2D method (2D slice-based) while preserving good detailed texture information, tended to underestimate the activities in organs and overestimate the neck and leg parts, due to the lack of spatial information during training. The conventional 3D method (3D volume-based) did better in recovering the activity in organs but rendered blurry and over-smoothed images.

We hypothesized that although NASC-PET images do not contain explicit information about photon attenuation and scattering, trained deep neural networks could predict low-frequency anatomy-dependent information which can be applied for correction. Our approach explores a simple way to incorporate domain knowledge, which can be intuitively understood as a domain (spatial) transformation. With such domain transformation, attenuation correction can be learned in a machine-independent low-frequency domain, while the original texture can be preserved in the processing. This can improve the robustness and effectiveness of learning to a large extent. Also, deep

learning is more transparently interpretable in the domain-transformed space. As shown in Fig. 5, motivated by the physical principle of ACF and scatter contribution[53], our proposed Decomposition-based DL modified one step of the conventional 3D method, i.e., we regularized the network to estimate low-spatial frequency anatomy-dependent information derived from NASC-PET, which is a voxel-wise ratio map. Detailed implementation would be described in the Method section. The comparison between Decomposition-based DL and conventional 3D DL can be viewed as an ablation study that demonstrated not only the advantages of our modification, but also the possibility of incorporating domain knowledge into the design of artificial intelligence in a simple way. Meanwhile, since we decomposed the complex end-to-end generation from NASC-PET to ASC-PET into two components, anatomy-independent textures (relating to tracers and diseases) and anatomy-dependent correction, as a result, we posed an easier learning task for the model so that we could increase both the efficiency and robustness. The robustness of our Decomposition-based DL was verified in tests of previously unseen imaging tracers and from different scanners. In addition, our design allows NASC-PET input at low resolution ($6.6 \times 6.6 \times 8$ mm/voxel), small matrix size ($112 \times 112 \times 112$), which are lower than almost all current scanners. Therefore, our method is insensitive to differences of matrix sizes or resolution and can be applied to different scanners. In the cross-scanner scenario, although we attempted to calibrate data from different centers and vendors based on phantoms, varieties still exist due to scanner properties (Supplementary Table S1) or normalization methods, among others. Consequently, the prediction accuracy of DMI and uMI 780 was not optimal compared to Vision 450 or Vision 600. Moreover, we also observed lower uptake at the upper edge of the liver in Vision 450 and uMI 780 imaging of CT ASC-PET (Fig. 3), possibly due to the local mismatch between NASC-PET and CT images caused by respiratory motion effects, while the results of all three DL-based approaches showed no such artifacts. On the cross-tracer setting, among all test groups except the native [18]F-FDG, [68]Ga-FAPI showed the best results, while [18]F-PSMA, [68]Ga-DOTA-TATE, and [68]Ga-DOTA-TOC were slightly inferior, which may be related to the different distribution of receptor-specific tracers in the body compared to [18]F-FDG. In particular, [68]Ga-DOTA-TATE and [68]Ga-DOTA-TOC have higher uptake in the liver and spleen, and [18]F-PSMA has higher uptake in the glands, while [68]Ga-FAPI has a similar level of uptake in normal organs. Furthermore, the large variability imposed in cross-tracer, cross-scanner, and cross-center ([68]Ga-DOTA-TOC and [18]F-PSMA on Vision 600) can place too much burden on the DL model trained with limited complexity. Additionally, as shown in Supplementary Fig. 6 and Supplementary Table S2, the performance of our model was not affected by the weight distribution of the subjects, as body mass index (BMI) was generally not associated with NRMSE.

Our study trained a model on a homogeneous dataset with only one scanner and one tracer, which is not optimal for the DL development. Meanwhile, the performance of our model is subject to NASC-PET, which implies that it will not perform well when dealing with extreme cases, such as patients with upside-down images, contaminated patients, patients with implants, or abnormally high activity in certain regions. Supplementary Fig. 5 showed two extreme cases where our model overestimated the activity in the neck region, probably due to the abnormally high activity of NASC-PET images. Data crafting[54] or unsupervised domain adaptation[55] may help with performance degradation on unseen images. Nevertheless, our preliminary results confirmed the potential of our initial hypothesis, albeit in such a challenging cross-scanner and cross-tracer setup. This proof-of-concept can therefore support the design of more realistic studies in the future, by including a larger and heterogeneous dataset that is not limited by the center, scanner, tracer, disease, or body region. Additionally, the proposed method needs further clinical validation in

clinical applications, especially clinical quantitative scores (refer to quantitative clinical scores such as the PERCIST[56] or the Deauville-Score[57], and so on).

## Methods

### Data preparation

This retrospective study complies with all relevant ethical regulations of the respective local ethics committees in Switzerland (Waiver from Cantonal Ethics Committee of Bern, Switzerland) and China (Approval from Ruijin Hospital Ethics Committee Shanghai Jiao Tong University School of Medicine). All patients included gave written informed general consent for retrospective analysis of their data. This is a retrospective study, eight cohorts with 829 subjects were included (Table 1). The proposed DL algorithm was developed based on 470 subjects data and the evaluation was performed using the unseen external datasets of different tracers and scanners.

A CT scan was performed prior to PET data acquisition for attenuation correction. Scatter correction was performed only on PET-CTAC images using a single-scatter simulation (SSS) algorithm[58] with two iterations. All data were reconstructed using Ordered Subsets-Expectation Maximization. More detailed information concerning scanner properties and reconstruction parameters can be found in Supplementary Table S1.

### Decomposition-based DL

We decomposed the complex end-to-end generation from NASC-PET to ASC-PET into two components, anatomy-independent textures (relating to tracers and diseases) and anatomy-dependent correction, and regularized a 3D deep neural network to estimate this low-frequency anatomy-dependent correction information only.

As shown in Fig. 5, we first calculated the anatomy-dependent correction map (ADCM) based on NASC-PET and ASC-PET according to the following equations:

$$\text{If } I^{NASC-PET}[x, y, z] > \varepsilon \text{ then}$$
$$I^{ADCM}[x, y, z] = \frac{I^{ASC-PET}[x, y, z]}{I^{NASC-PET}[x, y, z]} \quad (1)$$
$$\text{else } I^{ADCM}[x, y, z] = I^{ASC-PET}[x, y, z]$$

where we set $\varepsilon$ to be 1. To preserve more spatial information, which is most essential for the task of attenuation and scatter correction, we downsampled the NASC-PET and ADCM to a size of $112 \times 112 \times 112$. A semi-supervised 3D conditional generative adversarial network (c-GAN)[59] was employed, which consists of a generator network (G) to synthesize the ADCM from NASC-PET, and a discriminator (D) to distinguish between the synthesized ADCM and the real inputs, the objective function is defined as:

$$\min_G \max_D V(D, G) + \lambda V(G)$$
$$= \sum_{i=1}^{n} \log\left(D_{\theta_D}\left(I^{NASC-PET}, I^{ADCM}\right)\right)$$
$$+ \log\left(1 - D_{\theta_D}\left(I^{NASC-PET}, G_{\theta_G}\left(I^{NASC-PET}\right)\right)\right) \quad (2)$$
$$+ \lambda \left| G_{\theta_G}\left(I^{NASC-PET}\right) - I^{ADCM} \right|_2$$

where $\lambda$ is a weighting of loss, which was set to 1e+4 based on experiments. The model was trained on 470 3D images of size $112 \times 112 \times 112$ and validated on 51 reserved subjects (Table 1).

In the testing stage, given a NASC-PET, the trained generator network G was used to predict the DL-generated ADCM ($I^{DL-ADCM}$), and applied to NASC-PET to obtain DL ASC-PET according to the following equations:

$$\text{If } I^{NASC-PET}[x, y, z] > \varepsilon \text{ then}$$
$$I^{DL\ ASC-PET}[x, y, z] = I^{NASC-PET}[x, y, z] * I^{DL-ADCM}[x, y, z] \quad (3)$$
$$\text{else } I^{DL\ ASC-PET}[x, y, z] = I^{NASC-PET}[x, y, z]$$

To evaluate the quality of the DL ASC-PET images, we calculated the global physical metrics, to the CT ASC-PET, including voxel-wise NRMSE, PSNR, and SSIM. Furthermore, on six randomly selected subjects in each dataset, two certified nuclear medicine physicians (R.G. and C.M.) used ITK-SNAP to manually delineate three spherical VOIs within each target organ (liver, kidney, and heart). This was followed by SUV measurements as well as statistical analysis of clinical imaging parameters including $SUV_{mean}$, $SUV_{max}$, total lesion metabolism (TLM), as well as the most relevant radiomics features. The performance of our proposed Decomposition-based DL method was further investigated through comparison with traditional direct DL approaches, differences between each group for NRMSE were assessed for statistical significance using the paired two-tailed $t$-test, with a statistically significant difference defined as $p < 0.05$ with Bonferroni correction for minimizing type I error. More information on the data preprocessing, network design, training procedure, and physical metrics is attached in the corresponding part of the Supplementary material.

### Reporting summary

Further information on research design is available in the Nature Research Reporting Summary linked to this article.

## Data availability

The data that support the findings of this study are available from the corresponding author (BL) with the completion of data transfer agreement (DTA), and the request will be answered within thirty (30) days of receiving of the DTA.

## Code availability

Code for the Decomposition-based DL architecture is available at https://github.com/LeoXue09/CT-free-PET.

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

## Acknowledgements

This work was supported by the Swiss National Science Foundation (No. 188350, KS), National Natural Science Foundation of China (No. 82171975, R.G. and 82171971, B.L.) and Shanghai Municipal Key Clinical Specialty (No. shslczdzk03403, B.L.).

## Author contributions

K.S., B.L. and S.X. conceived and designed the study. R.G., Y.W., and Y.Z. screened and collected clinical information and imaging data. S.X. conceived and carried out data analysis. R.G. reviewed all analysis. S.X., R.G., and K.S. drafted the manuscript. J.H., H.S., C.M., K.Z., G.P., M.V., R.S. and A.R. reviewed and revised the manuscript for important intellectual content.

## Competing interests

H.S. is a full-time employee of Siemens Healthineers AG in Switzerland. K.S. and A.R. have received research grants from Siemens Healthineers AG and Novartis AG. The remaining authors declare no competing interests.
