## [Peer Review File · Nature Communications]

Reviewers' Comments:

Reviewer #1:

Remarks to the Author:

Practice of domain knowledge in trustworthy deep learning for CT-free PET imaging

The manuscript presents a DL method based on GANs to integrate domain knowledge in PET imaging for attenuation correction without the use of CT. They trained a base model on images from a specific scanner by one vendor and the tracer FDG and showed that their method yielded high quality and quantitative accuracy attenuation corrected PET images from other scanners and other tracers. The topic is relevant and interesting and helps motivate the use of DL methods in the clinic. PET imaging being very diverse, these kind of studies are important. There are a lot of details and motivations missing in the methods section however that needs to be included.

1. LINE 53: You state that a considerable dose contribution comes from the CT in PET/CT. Is this really true, since PET/CT almost always use low-dose CT? You later comment on low-dose CT, but this first statement may not be entirely accurate.
2. LINE 68: Explain cross-talk artifact briefly.
3. LINE 115: Describe the statistical test performed. Did you correct for multiple comparisons?
4. It seems the conventional 2D method is better than the conventional 3D method (both worse than the decomposition method however). Please comment on this? Why would 2D be better than 3D?
5. LINE 124: How were the organ VOIs annotated? Manually? By whom? Explain.
6. FIGURE 2: It would have been useful to see difference images as well, which makes patterns more clear. For example, it seems your proposed method often overestimates the activity in the neck region (especially obvious in col 1, bottom row and col 2, top row). Please comment on this, if this is generally true, why this happens, and what the clinical implications are. E.g. is your method not suitable for certain areas?
7. The test subjects with different tracers (not FDG) were scanned on two different scanner models, Siemens 450 (same as training), and Siemens 600. Additionally, DOTATOC and PSMA were scanned at different institutions (Bern) compared to the training set (SH). How did you separate the effects of the different tracer v. different scanner v. different institution for the different tracer experiments? Eg. you have results for FDG Siemens 600 from Bern to compare to training FDG Siemens 450 SH, but it's not clear if you've corrected for this difference in the tracer experiments with DOTATOC and PSMA (Siemens 600 Bern).
8. While it is ok to place some of the method in the supplementary material, there needs to be more detail in the main manuscript. The neural network design section should be in the main manuscript with more detail.
9. There is a lot of information missing in the methods (even supplementary material). What were the network filter sizes? Were the images pre-processed in any way (normalization, resampling, ...)? What were the input image sizes? Did you use batch normalization? Is your decomposition model a full 3D model, or 2D slice by slice?
10. How were hyperparameters tuned? It is customary to use a validation set. Did you? There is a column "valid" in table 1, but no real explanation. How many epochs did you train, and did you use early stopping?

Minor comments

11. LINE 61: The paragraph "...such as avoid..." is unclear and hard to understand. Please rewrite.
12. Figure 2 is referenced after figure 3 in the text.
13. LINE 118: The wording "accuracy" refers to NRMSE here? Confusing to say accuracy, please rewrite.
14. FIGURE 4: The text in the figure is much too small to be legible. Enlarge.
15. LINE 173: "PSMA, while".
16. LINE 224: "increasing".
17. LINE 257: "vendor" instead of "merchandise"
18. LINE 504: "image".
19. SUPP FIGURE 1: Text and lines too small to be legible.

Reviewer #2:

Remarks to the Author:

The manuscript was rejected due to the following major issues:

1. Abstract and Introduction "This study employs a simple way to integrate domain knowledge in deep learning for CT-free PET imaging." However, the domain knowledge does not appear in the network design. It is just verified by some experimental results.
2. Abstract and Introduction "we decomposed the complex end-to-end generation into two components, anatomy-independent textures (relating to tracers and diseases) and anatomy-dependent correction." However, the former does not appear in the Methods part. What role does it play in the network design?
3. Methods "As shown in Figure 5, we first calculated the anatomy-dependent correction map (ADCM) based on NASC-PET and ASC-PET, which is a voxel wise ratio map." What are the differences between ADCM and attenuation maps (μ -maps)? How to calculate the ADCM?
4. Although the proposed network performs well in this paper, it is inherently similar to those direct approaches (conversion from NASC PET to ASC PET). More creativity is desired in the work. Besides, there are also some details in the manuscript that could be improved:
 1. Introduction, regarding the MLAA algorithm, some improved methods should be mentioned, such as MLAA+kernel proposed by Guobao Wang et al. The improved methods may alleviate the low-frequency crosstalk problem.
 2. Discussion, at the end of the first paragraph, the role or influence of CT scanning was not explained in the example of drug development. If this example is used to illustrate that CT-free PET imaging is beneficial in this situation, more explanation should be supplemented on why CT scans should be reduced, or even no CT scans.

Reviewer #3:

Remarks to the Author:

In their manuscript "Practice of domain knowledge in trustworthy deep learning for CT-free PET imaging" the authors describe a method for deep learning-based CT-free PET attenuation correction.

The general topic of this work - enabling PET image reconstruction and attenuation/scatter correction without the necessity of CT data - is of general relevance to the field. The specific challenge that the authors address - improving robustness and generalizability of related methods across different scanners and tracers - is clinically and scientifically relevant.

The authors provide results on a large and representative clinical PET dataset including a variety of scanners and PET tracers.

However, this manuscript has major and minor shortcomings that need to be addressed:

Major points:

1) In general, the methodology of this work is not described in sufficient detail in order for the reader to understand important details.

Importantly, the central aspect of this work - the approach of decomposing the attenuation correction step into two separate parts - is not defined accurately.

Based on the descriptions in the manuscript I assume that the authors estimate a voxelwise ratio map between attenuation-corrected and non-attenuation-corrected PET images.

This core idea should be defined unambiguously using a clear mathematical description (which ratio exactly, how are areas of zero or very low uptake treated etc.).

Also in this context, it is unclear how this method should separate "low-frequency" and "high-frequency" components of the image.

2) If my understanding of the proposed method is correct, it remains unclear to me why this method should in principle be more robust and generalizable than the direct estimation of corrected PET images. The only conceptual difference this direct estimation is a different

representation of the estimation target (a ratio map), which can deterministically and reversibly be translated to the conventional estimation target (the corrected PET image). Thus, it is not evident at which step "domain knowledge" is used as the authors claim.

The authors should motivate and explain why their proposed method can be expected to show better performance and whether also other aspects (e.g. data preprocessing, data augmentation, network architecture, training process) are different compared to the baseline methods. Additional ablation studies could provide this insight.

3) The proposed method of estimating a pixel-wise ration map can be interpreted as projecting the integrated attenuation of tissue around each voxel within each voxel. Increased robustness of the proposed method might therefore stem from the fact that only a rough estimate of surrounding tissue mass might be required to estimate this ration map, which might be an easier task to learn. Potentially, the described method might however deteriorate in unusually obese or cachectic patients. This should be investigated e.g. by plotting BMI-dependent quantification errors (the relatively inferior performance on the UI and GE scanners might be a hint in this direction as the respective populations have significantly different weight distributions compared to the training population).

4) The approach proposed by the authors is somewhat similar to an alternative approach of estimating a pseudo-CT as an intermediate step. This alternative approach however actually adds prior knowledge as it represents the physical process of PET correction and allows for subsequent physics-based image reconstruction and correction. Therefore, this approach (pseudo-CT as an intermediate step) should be included as a baseline of comparison.

5) The approach for quantitative evaluation of algorithm accuracy is rather intransparent due to the following reasons:

- The most important metric for comparing methods of PET attenuation correction should be voxel-based activity measures (either as Bq/ml or SUV). Texture features / similarity metrics are of only secondary relevance to the task of PET attenuation/scatter correction.
- Quantitative PET values should be measured in different organs (e.g. using regions of interest) in order to provide a clear picture of the spatial distribution of quantification errors and potential anatomical variability.
- The errors should be provided as mean absolute errors (not as mean errors as these can cancel out) over each region of interest. The chosen metrics do not provide this level of detail.
- Instead of providing scatter plots, it would be much more informative to provide voxel-wise maps of the absolute error for representative examples as this would again give insight into the spatial anatomical distribution of errors
- The authors do not describe how lesions were identified to measure related metrics: how many lesions were identified, how were they segmented, where were they located etc.

Further points:

6) Title: The title should be more specific. E.g. it is unclear what the term "trustworthy" means in this context. One suggestion might be: "Towards robust and generalizable machine learning-based CT-free PET attenuation correction"

7) Abstract: The core idea of the method should be briefly described in the abstract. Similar to the rest of the manuscript it remains unclear what the described "decomposition" actually consists of.

8) Introduction: The authors motivate their method in part by referring to total-body PET/CT scanners. However, it should be mentioned in this context that alternative approaches for CT-free PET correction are being developed for these scanners that rely on physical effects (e.g. https://jnm.snmjournals.org/content/62/supplement_1/1530)

9) In general the term AI (artificial intelligence) is rather vague. It seems preferable to use the term machine learning in the context of this work.

10) Results: see above. The level of detail regarding used metrics and performed analysis is not sufficient for valid assessment of the proposed method.

11) Discussion: Mentioned limitations should be addressed. Also, alternative methods for increasing algorithm robustness should be discussed.

12) Methods: see above. The level of detail regarding the methods (specifically also the proposed algorithm itself) is not sufficient in order for the reader to understand and potentially reproduce this work. Also, evaluation metrics need to be described more precisely.

13) Methods: It is to be expected that the "AI-generated anatomy dependent correction map" does not capture background noise (as also seen in Figure 5). This should result in high image background noise levels in the AI-corrected PET due to existing noise in the non-corrected PET. Is this the case? If not why?

This effect might also be relevant for low-count areas in the foreground. This should be further explained.

14) Methods: The different scanners have different image characteristics (e.g. matrix size). How is this accounted for at test time? Are images resampled to similar spatial resolutions?

Response to Review Comments on NCOMMS-22-03233

Response: We would like to thank again the Editors and Reviewers for their time and effort on our manuscript and for allowing us to revise this manuscript. Their valuable and constructive comments helped us to improve this study. Following the comments, we have modified the manuscript and have answered the questions carefully. In particular, we:

1. Modified the Method section with a more detailed description and added more information on data preprocessing, network design, training procedure, and physical metrics in Supplementary material, to increase the readability and reproducibility.
2. Implemented several state-of-art attenuation correction methods (Pseudo-CT based, MLAA) and compared the results to our proposed Decomposition-based deep learning (DL) method.
3. Discussed in detail the way in which domain knowledge is incorporated, as well as the creativity and superiority of our approach compared to traditional AI approaches and its possible reasons.
4. Added more patient data for test including extreme cases, and improved the data analysis and result presentation.

We hope that the revision will allow the manuscript to be accepted for publication.

Comments from Reviewers:

Reviewer #1:

1. The manuscript presents a DL method based on GANs to integrate domain knowledge in PET imaging for attenuation correction without the use of CT. They trained a base model on images from a specific scanner by one vendor and the tracer FDG and showed that their method yielded high quality and quantitative accuracy attenuation corrected PET images from other scanners and other tracers. The topic is relevant and interesting and helps motivate the use of DL methods in the clinic. PET imaging being very diverse, these kinds of studies are important. There are a lot of details and motivations missing in the methods section however that needs to be included.

Response: We appreciate the Reviewer for the generally positive assessment and encouragement of our study. We added the missing details in this revision according to the constructive comments.

LINE 53: You state that a considerable dose contribution comes from the CT in PET/CT. Is this really true, since PET/CT almost always use low-dose CT? You later comment on low-dose CT, but this first statement may not be entirely accurate.

Response: Thanks for the Reviewer pointing out the insufficient statement. We agree that this is the most important motivation for the current study and we need to make the statement more carefully reflecting the current state of the art. Following the Reviewer's comment, we summarized the following points to clarify the motivation of this study:

1. The effective dose of CT for PET/CT imaging generally ranges from 1 mSv to 20 mSv depending on the purpose of CT ¹. In general, low dose attenuation correction CT is often recommended in the current practice of PET/CT imaging according to the EANM guidelines ². Although the effective dose of low-dose AC CT is much less than diagnostic CT (11.81 mSv ³), the effective dose of typical whole-body low-dose CT is still 6.4 mSv ⁴. The effective dose of 25% quantile of the diagnostic reference value of low dose AC CT in Switzerland is 3 mSv for trunk PET/CT imaging ⁵. The recent development of ultra-low dose attenuation correction CT (AC CT) imaging further reduces the radiation exposure to a median whole-body dose of 2.1 mSv ⁴. The dose of these low-dose AC CT is usually not negligible in the current PET/CT imaging practice, where the effective dose of PET is approximately 4.8 mSv¹ for widely applied FDG PET according to the current diagnostic reference value of EANM. In regional PET/CT applications the effective dose of low-dose AC CT can be less than 1 mSv. For example, the effective dose of AC CT for brain PET/CT imaging can be approximately 0.15 mSv ⁵. Only in these limited situations, the radiation burden of AC CT might be negligible.
2. In general, the current development does not aim to replace diagnostic CT in PET/CT imaging, although the radiation dose is considerably high. For those regional PET/CT imaging (e.g. brain) with very small radiation burden, the benefit of the current development is limited. However, avoiding AC CT, i.e. CT-free PET imaging, could be beneficial in several situations:
 - i. Often the patients referred for PET/CT imaging have previous diagnostic CT ⁶. Even ultra-low dose AC CT could be redundant and brings additional radiation burden for these patients, especially for pediatric patients ⁷.
 - ii. Some patients may have to go through a series of PET scans, the accumulated dose of multiple low-dose AC CT could be concern. For example, some patients require a series of ⁶⁸Ga-DOTATATE and ¹⁸F-FDG imaging for the diagnosis of neuroendocrine tumor ⁸, sometimes more than two tracers are used for differential diagnosis ⁹⁻¹¹. However, repeated AC CT in a short period of time doesn't bring new information but increases the radiation dose.
 - iii. For pharmaceutical developments, multiple low-dose AC CT may be redundant.

For example, multiple PET/CT is often needed for ^{89}Zr antibody imaging ^{12,13}, where avoiding repeating low-dose AC CT could be beneficial.

3. The advancement of technologies continuously reduces the radiation burden of both PET and CT imaging. The latest large axial field of view (LAFOV) PET scanners can reduce the PET dose potentially to less than 1 mSv ¹⁴. Despite the efforts of reducing the radiation burden of CT, the low dose AC CT could still be a considerable part of the dose in advanced PET/CT systems.

Following the suggestion, we added these clarifications in the revision.

2. *LINE 68: Explain cross-talk artifact briefly.*

Response: Thanks for the Reviewer's suggestion. We agree that simply mentioning cross-talk artifact ^{15,16} is not self-explanatory enough. Therefore, we have modified this sentence on Page 3 and added two updated MLAA-based methods that are intended to mitigate the effects of this cross-crosstalk: ““Alternatively, attenuation maps (μ -maps) can also be derived using the maximum likelihood estimation of activity and attenuation (MLAA) algorithm. MLAA can be further improved with the use of the additional time of flight (TOF) ^{17,18}. Due to the insufficient timing resolution of current clinical PET systems, the MLAA suffers from the crosstalk between the activity and attenuation distribution and high noise ¹⁹. Recent MLAA methods alleviate the low-frequency crosstalk problem ²⁰, but may over-smooth or over-estimate the bone structure ^{21,22}. TOF-MLAA-based approaches using an external source like rod ¹⁷ or lutetium oxyorthosilicate background transmission ²³ are under development.”

Furthermore, we've compared our Decomposition-based method to a TOF-MLAA-based approach ¹⁶ on data scanned with Siemens Biograph Vision Quadra. The preliminary results (Figure 10) showed that our method performed well on Quadra both quantitatively and qualitatively, while the MLAA-based method tended to overestimate in the skin and underestimate in the bladder.

3. *LINE 115: Describe the statistical test performed. Did you correct for multiple comparisons?*

Response: Thanks for the kind reminder from the Reviewer's comments. We've added the information about the way how we compared the accuracy of our method to the other conventional AI methods in the Method part on page 9: “The performance of our proposed Decomposition-based DL method was further investigated through comparison with traditional direct AI approaches, differences between each group for NRMSE were assessed for statistical significance by means of the paired two-tailed t-test, with a statistically significant difference defined as $p < 0.05$ with Bonferroni correction for minimizing type I error.”

Following the suggestion of correction for the multiple comparisons, we performed

Bonferroni correction in the revision, and the results are updated accordingly on Page 4&5 as follows: “As shown in Figure 1. A-C, all three AI methods were capable of some degree of attenuation and scattering correction for different scanners, but Decomposition-based DL significantly outperformed the other two on all scanners ($p<0.025$).” “All three AI methods were capable of some degree of attenuation and scattering correction for different tracers, but Decomposition-based DL significantly outperformed the other two on ^{68}Ga -FAPI and ^{18}F -PSMA ($p<0.025$).”

4. *It seems the conventional 2D method is better than the conventional 3D method (both worse than the decomposition method however). Please comment on this? Why would 2D be better than 3D?*

Response: Thanks for the Reviewer’s comments. Theoretically, 3D method should perform better than the 2D method within direct approach. However, it is still not possible to train whole body data in 3D method with the typical computational resources of most research institutes. To solve this problem, we can train the network either with downsampled whole-body data or patch-based data. Previously we only did the downsampling to make it consistent with our decomposition-based method, where the results are blurry for the 3D-based method. Following your suggestion, we also did 3D patch-based training. The results are shown in Figure 1.

Figure 1. Comparison between different AI approaches, including newly implemented 3D patch-based.

The results of physical metrics and visual readings showed that our Decomposition-based DL performed the best and 2D slice-based method performed still better than either 3D patched-based or 3D volume-based AI.

For proper attenuation correction, global anatomical information is critical for AI-based

methods. Ideally, the intact 3D volume should be taken as input, which is not feasible with the limited GPU capacity in typical research institutes. In our decomposition-based method, we are able to down-sample the learning data without affecting the overall resolution. For conventional methods, we can only compromise it and tried to consider as much context anatomy as possible in the learning procedure. Compared to the limited global information in 3D patches, the 2D slice can have at least whole global anatomical information in the axial plane. Therefore, the 2D AI method performed better than 3D patch-based method. On the other side, the conventional 3D volume-based AI method has to be down-sampled and leads to resolution degradation. The decomposition of our method allows global learning and resolution preservation at the same time and consequently achieves better image quality, preserving better detailed information as well as spatial information.

5. *LINE 124: How were the organ VOIs annotated? Manually? By whom? Explain.*

Response: Thanks for the kind reminder from the Reviewer’s comments. We added this missing information on Page 8: “Furthermore, on six randomly selected subjects in each dataset, two certified nuclear medicine physicians (R.G. and C.M.) used ITK-SNAP to manually delineate three spherical VOIs within each target organ (liver, kidney, and heart).” We include this VOI delineation in the demo data as well, as shown in Figure 2.

Figure 2. Snapshot demo of spherical VOI delineation.

Following the suggestion, we considered more ROIs to avoid biases in the revision.

We added 3 VOIs for each organ and updated the results in Figure 3.

Figure 3. Quantitative accuracy of the DL ASC-PET images generated with our proposed method, to the CT ASC-PET on the cross-scanner and cross-tracer setting, evaluated with clinical imaging parameters such as SUV_{mean} , SUV_{max} , total lesion glycolysis (TLG), as well as the most relevant radiomics features. (Figure 2 in manuscript)

The updated results showed still that Decomposition-based AI outperformed the other two on all scanners and all tracers regarding local metrics, only that all groups had slightly higher error rates compared to previous results.

6. *FIGURE 2: It would have been useful to see difference images as well, which makes patterns more clear. For example, it seems your proposed method often overestimates the activity in the neck region (especially obvious in col 1, bottom row and col 2, top row). Please comment on this, if this is generally true, why this happens, and what the clinical implications are. E.g. is your method not suitable for certain areas?*

Response: Thanks for the suggestion from the reviewer, we agree that plotting the difference image would make the pattern clearer, so we added Figure 4 to the Supplementary material.

Figure 4. Voxel-wise absolute percentage error map of an exemplary subject depicting the difference between DL ASC-PET and reference CT ASC-PET for different scanners and tracers. (**Supplementary FIGURE 3**).

Following your suggestion, we re-investigated the results from the two datasets you referred to, GE Discovery MI (*col 1, bottom row*) and ^{68}Ga -FAPI (*col 2, top row*). This overestimation in the neck region happened only on a few subjects (3/104 in GE, 1/7 in FAPI). As shown in Figure 5, we randomly picked two more subjects and they do not have such a problem.

Figure 5. Randomly picked two more subjects from the dataset of GE Discovery MI and ^{68}Ga -FAPI, that do not show overestimation in the neck region.

Therefore, we updated the results with subjects without such an overestimation problem (shown in Figure 6). We discussed these extreme cases in the Discussion and showed the images in the supplementary material (Figure 7). Otherwise, it might be misleading for the readers. As shown in Figure 7, the reason caused the overestimation might be related to the abnormally high activity of non-corrected images in the neck region.

Figure 6. Exemplary test results of ^{18}F -FDG imaging from SIEMENS Healthineers Vision 450 (SH), Vision 600 (Bern), UI uMI 780 (SH) and GE Discovery MI (SH) and imaging from ^{68}Ga -FAPI, ^{68}Ga -DOTA-TATE (Vision 450-SH), ^{68}Ga -DOTA-TOC and ^{18}F -PSMA (Vision 600-Bern). Note that the color bars used for non corrected PET are in the range of 15% of the presented color bars. (Figure 3 in manuscript)

Figure 7. Cases showing overestimation in the neck region. (Supplementary FIGURE 4).

7. The test subjects with different tracers (not FDG) were scanned on two different scanner models, Siemens 450 (same as training), and Siemens 600. Additionally, DOTATOC and PSMA were scanned at different institutions (Bern) compared to the training set (SH). How did you separate the effects of the different tracer v. different scanner v. different institution for the different tracer experiments? Eg. you have results for FDG Siemens 600 from Bern to compare to training FDG Siemens 450 SH, but it's not clear if you've corrected for this difference in the tracer experiments with DOTATOC and PSMA (Siemens 600 Bern).

Response: Thanks for the Reviewer's comments. We totally agree that it is quite important to separate the effects of the different tracers, different scanners and different institutions, which would help us and the reader to learn more about the advantage and limitation of our method. We would like to include this work in our future plan, by conducting a prospective study and acquiring a control group and experiment group with only one difference mentioned above. And for our current retrospective study, this

is indeed one of our limitations and imposed more complexity on our model when applying to such a dataset with both cross-scanner and cross-tracer differences, e.g. ^{68}Ga -DOTATOC and ^{18}F -PSMA on Vision 600. Hence we further discussed this effect on Page 8 as follows: “Furthermore, the large variability imposed in cross-tracer, cross-scanner and cross-center (^{68}Ga -DOTATOC and ^{18}F -PSMA on Vision 600) can place too much burden on the AI model trained with limited complexity.”

8. While it is ok to place some of the method in the supplementary material, there needs to be more detail in the main manuscript. The neural network design section should be in the main manuscript with more detail.

Response: Thanks for the suggestion from the reviewer. We totally agree that we should describe our method in more detail, so we modified the **Method** section as shown in the “Modified Method section in Manuscript” attached after reference to this response letter. We’ve also added a Figure describing our network architecture in the Supplementary materials as shown in Figure 8.

Figure 8. Network architecture. (Supplementary FIGURE 1).

9. There is a lot of information missing in the methods (even supplementary material). What were the network filter sizes? Were the images pre-processed in any way (normalization, resampling, ...)? What were the input image sizes? Did you use batch normalization? Is your decomposition model a full 3D model, or 2D slice by slice?

Response: Thanks for the Reviewer’s comments. We agree that including more detailed information would improve the reproducibility of our project. Here are the details: Our network is a full 3D network. We used 3×3×3 filters, applied min-max normalization, and downsampled the image to 112×112×112 as input to the network. Batch normalization was applied in the discriminator, which made the training more efficient, while the generator was used to synthesize images with absolute values so

that batch normalization was not applied. We added all this information and others in Supplementary materials as shown in the “Modified Method section in Supplementary materials” attached after reference of this response letter.

10. *How were hyperparameters tuned? It is customary to use a validation set. Did you? There is a column “valid” in table 1, but no real explanation. How many epochs did you train, and did you use early stopping?*

Response: Thanks for the Reviewer’s comments. Our model was trained on 470 subjects and validated on 51 reserved subjects to tune the hyperparameters. The training process ran for 100 epochs and no early stopping was used. We added all this information in either the manuscript or the Supplementary material as indicated in the answer above.

11. *LINE 61: The paragraph “...such as avoid...” is unclear and hard to understand. Please rewrite.*

Response: Thanks for the Reviewer’s comments. We modified this sentence as follows on Page 2: “eliminating AC CT, i.e. CT-free PET imaging, is beneficial in a number of situations such as pediatric examinations, multiple PET/CT examinations, pharmaceutical tests, and so on, where additional radiation burden due to redundant anatomic imaging could be concerned.”

12. *Figure 2 is referenced after figure 3 in the text.*

Response: Thanks to the reviewer for your kind reminders. We’ve changed the order of Figure 2 and Figure 3.

13. *LINE 118: The wording “accuracy” refers to NRMSE here? Confusing to say accuracy, please rewrite.*

Response: Thanks for the Reviewer’s comments. We modified this sentence on Page 4 as follows: “Specifically in terms of normalized root mean squared error (NRMSE), Decomposition-based DL improved 47.5% over Direct 2D and 49.1% over Direct 3D on Vision 450, and 60.0% over Direct 2D and 58.4% over Direct 3D on Vision 600, while on both scanners maintained a similar level of error ($p=0.88$).”

14. *FIGURE 4: The text in the figure is much too small to be legible. Enlarge.*

Response: Thanks for the suggestion from the reviewer. We enlarged the text and also split Figure 4 into two parts so that the text is more readable.

15. *LINE 173: “PSMA, while”. LINE 224: “increasing”. LINE 257: “vendor” instead of “merchandise”. LINE 504: “image”.*

Response: Thanks to the reviewer for your kind reminders. We’ve corrected all mentioned typos or mistakes.

16. *SUPP FIGURE 1: Text and lines too small to be legible.*

Response: Thanks for the Reviewer’s comments. We split Supplementary Figure 1. to two parts so that the text is more readable.

Reviewer #2:

1. *The manuscript was rejected due to the following major issues:*

Abstract and Introduction “This study employs a simple way to integrate domain knowledge in deep learning for CT-free PET imaging.” However, the domain knowledge does not appear in the network design. It is just verified by some experimental results.

Response: Thanks for the Reviewer’s comments. We agree that it could be mathematically beautiful if the domain knowledge can be directly integrated into the deep neural network, e.g. the loss function^{24,25} or the network architecture²⁶. In contrast, to learn the domain knowledge with complex mathematics, our method explored a simple way for domain knowledge integration. The domain knowledge integration can be intuitively understood as similar to domain (space) transformation, which can be derived from practical knowledge in PET imaging. With this domain transformation, the attenuation correction can be learned in a low-frequency machine-independent domain while the original texture can be preserved in the processing. This can largely enhance the robustness and effectiveness of the learning. Also, deep learning is more transparent and interpretable in the domain-transformed space.

We agree that we should explicitly state the domain knowledge we incorporated here, as well as the advantage of our approach compared to the conventional AI method. So we modified the Discussion section as follows: “We hypothesized that although NASC-PET images do not contain explicit information about photon attenuation and scattering, trained deep neural networks could predict low-frequency anatomy-dependent information which can be applied for correction. We hypothesized that although NASC-PET images do not contain explicit information about photon attenuation and scattering, trained deep neural networks could predict low-frequency anatomy-dependent information which can be applied for correction. Our approach explores a simple way to incorporate domain knowledge, which can be intuitively understood as a domain (spatial) transformation. With such domain transformation, attenuation correction can be learned in a machine-independent low-frequency domain, while the original texture can be preserved in the processing. This can improve the robustness and effectiveness of learning to a large extent. Also, deep learning is more transparently interpretable in the domain-transformed space. As shown in Figure 5, motivated by the physical principle of attenuation correction factors (ACF) and scatter contribution, our proposed Decomposition-based DL modified one step of the conventional 3D method, i.e., we

regularized the network to estimate low-spatial frequency anatomy-dependent information derived from NASC-PET, which is a voxel-wise ratio map. Detailed implementation would be described in the Method section. The comparison between Decomposition-based DL and conventional 3D AI can be viewed as an ablation study that demonstrated not only the advantages of our modification but also the possibility of incorporating domain knowledge into the design of artificial intelligence in a simple way. Meanwhile, since we decomposed the complex end-to-end generation from NASC-PET to ASC-PET into two components, anatomy-independent textures (relating to tracers and diseases) and anatomy-dependent correction, as a result, we posed an easier learning task for the model so that we could increase both the efficiency and robustness.”

Meanwhile, we agree that our previous description of our method is rather intransparent, so we modified the **Method** section as shown in the “Modified Method section in Manuscript” attached after the Reference of this response letter.

2. *Abstract and Introduction* “we decomposed the complex end-to-end generation into two components, anatomy-independent textures (relating to tracers and diseases) and anatomy-dependent correction.” However, the former does not appear in the *Methods* part. What role does it plays in the network design?

Response: Thanks for the Reviewer’s suggestion. We agree that the description of our proposed method should be stated clearer in the Abstract, so we modified the sentence as follows: “This study employs a simple way to integrate domain knowledge in deep learning for CT-free PET imaging. In contrast to conventional direct deep learning methods, we regularized a 3D deep neural network to estimate attenuation and scatter correction information in a machine-independent low-frequency domain. The efficiency and robustness of our proposed approach were verified in tests of external imaging tracers on different scanners.” We added detailed information in the Method part as indicated in the answer above.

3. *Methods* “As shown in Figure 5, we first calculated the anatomy-dependent correction map (ADCM) based on NASC-PET and ASC-PET, which is a voxel wise ratio map.” What are the differences between ADCM and attenuation maps (μ -maps)? How to calculate the ADCM?

Response: Thanks for the Reviewer’s suggestions. Attenuation-correction factors (ACF) calculated from μ -map are voxel-wise ratios to be applied to NASC-PET, to compensate for the loss of detected photon pairs. Our proposed ADCM is also a voxel-wise ratio map calculated from NASC-PET and ASC-PET, which potentially includes information about photon attenuation and scattering.

Here’s an example of the ADCM and CT-based μ -map shown in Figure 9.

Figure 9. Example of the anatomy-dependent correction map (ADCM) and CT-based μ -map.

We also included more information about the ADCM in Supplementary material, as shown in the “Modified Method section in Supplementary materials” attached after reference of this response letter.

4. *Although the proposed network performs well in this paper, it is inherently similar to those direct approaches (conversion from NASC PET to ASC PET). More creativity is desired in the work.*

Response: Thanks for the kind reminder from the Reviewer’s comments. We agree that our deep neural network structure is similar as direct approaches. In contrast to developing a more complicated deep neural network structure. We developed a decomposition-based method, which transfers the problem into another domain, which can be better and more robustly learned, and the results are then transferred back to the original domain. This domain adaption is derived from practical experience and the simple implementation is more robust in practice. Our primary concern about such direct approaches is that without well-designed regularization and penalty system, the predictions could overfit the training data and may not be consistent with the governing physical laws. And the comparison between our Decomposition-based DL and conventional 3D AI can be viewed as an ablation study that demonstrated not only the advantages of our modification but also the possibility of incorporating domain knowledge into the design of artificial intelligence in a simple way.

5. *Besides, there are also some details in the manuscript that could be improved:*

Introduction, regarding the MLAA algorithm, some improved methods should be mentioned, such as MLAA+kenel proposed by Guobao Wang et al. The improved

methods may alleviate the low-frequency crosstalk problem.

Response: Thanks for the Reviewer’s suggestion. Following the suggestion, we updated the statement on Page 3 with new MLAA-based methods that are intended to mitigate the effects of this crosstalk, as well as other CT-less PET corrections based on other physical effects: “Therefore, numerous research efforts have been devoted to developing methods for CT-free PET correction. Magnetic resonance (MR)-based approaches yielded satisfactory results in brain PET ²⁷. Alternatively, attenuation maps (μ -maps) can also be derived using the maximum likelihood estimation of activity and attenuation (MLAA) algorithm. MLAA can be further improved with the use of the additional time of flight (TOF) ^{17,18}. Due to the insufficient timing resolution of current clinical PET systems, the MLAA suffers from the crosstalk between the activity and attenuation distribution and high noise ¹⁹. Recent MLAA methods alleviate the low-frequency crosstalk problem ²⁰, but may over-smooth or over-estimate the bone structure ^{21,22}. TOF-MLAA-based approaches using an external source like rod ¹⁷ or lutetium oxyorthosilicate background transmission ²³ are under development to overcome the limitation.”

Inspired by this comment, we’ve compared our Decomposition-based DL to one of the TOF-MLAA-based approaches ¹⁶ on data scanned with Siemens Biograph Vision Quadra. The preliminary results (Figure 10) showed that our method performed well on Quadra both quantitatively and qualitatively, while the MLAA-based method tended to overestimate in the skin and underestimate in the bladder, which was caused by the cross-talk problem. The LSO-based MLAA approach we implemented is a vendor prototype, which showed the common cross-talk problem, which can be alleviated by Guobao Wang’s kernel MLAA approach.

Figure 10. Comparison between our Decomposition-based method with a TOF-MLAA-based approach on data scanned with Siemens Biograph Vision Quadra ^{18}F -FDG.

6. *Discussion, at the end of the first paragraph, the role or influence of CT scanning was not explained in the example of drug development. If this example is used to illustrate that CT-free PET imaging is beneficial in this situation, more explanation should be supplemented on why CT scans should be reduced, or even no CT scans.*

Response: Thanks for the suggestion from the reviewer. We agree that this statement is rather vague without further explanation. In the development of radiopharmacies, for example, ^{89}Zr -labelled antibodies, a series of PET/CT scans are usually performed within 2 weeks after the injection²⁸. Repeated AC CT scans could be critical concerns in such kinds of investigations^{13,29}.

Hence, we modified this sentence on Page 6 as follows: “In the cases such as the development of ^{89}Zr -labelled antibodies PET imaging, a series of PET/CT scans are usually performed within 2 weeks after the injection. Repeated AC CT scans could be critical concerns in such kinds of investigations”

Reviewer #3:

1. *In their manuscript "Practice of domain knowledge in trustworthy deep learning*

for CT-free PET imaging" the authors describe a method for deep learning-based CT-free PET attenuation correction. The general topic of this work - enabling PET image reconstruction and attenuation/scatter correction without the necessity of CT data - is of general relevance to the field. The specific challenge that the authors address - improving robustness and generalizability of related methods across different scanners and tracers - is clinically and scientifically relevant. The authors provide results on a large and representative clinical PET dataset including a variety of scanners and PET tracers. However, this manuscript has major and minor shortcomings that need to be addressed:

In general, the methodology of this work is not described in sufficient detail in order for the reader to understand important details. Importantly, the central aspect of this work - the approach of decomposing the attenuation correction step into two separate parts - is not defined accurately. Based on the descriptions in the manuscript I assume that the authors estimate a voxelwise ratio map between attenuation-corrected and non-attenuation-corrected PET images. This core idea should be defined unambiguously using a clear mathematical description (which ration exactly, how are areas of zero or very low uptake treated etc.). Also in this context, it is unclear how this method should separate "low-frequency" and "high-frequency" components of the image.

Response: Thanks for the suggestion from the reviewer. We completely agree that we should describe our method in more detail, so we modified the **Method** section as shown in the "Modified Method section in Manuscript" attached after reference to this response letter.

We only mentioned a general protocol of this calculation in the manuscript, and we described it in more detail in the Supplementary materials, including the "how are areas of zero or very low uptake treated", as shown in the "Modified Method section in Supplementary materials" attached after reference of this response letter, we made all voxel value below 1 in NASC-PET to be 1.

2. *If my understanding of the proposed method is correct, it remains unclear to me why this method should in principle be more robust and generalizable than the direct estimation of corrected PET images. The only conceptual difference this direct estimation is a different representation of the estimation target (a ratio map), which can deterministically and reversibly be translated to the conventional estimation target (the corrected PET image). Thus, it is not evident at which step "domain knowledge" is used as the authors claim. The authors should motivate and explain why their proposed method can be expected to show better performance and whether also other aspects (e.g. data preprocessing, data augmentation, network architecture, training process) are different compared to the baseline methods. Additional ablation studies could provide this insight.*

Response: Thanks for the Reviewer's suggestion. We agree that we should explicitly

state the domain knowledge we incorporated here, as well as the advantage of our approach compared to the conventional DL method. As shown in Figure 11. Intuitively, we imposed an easier task for the neural network to learn, compared to conventional DL, we regularized the network to estimate only low-spatial frequency anatomy-dependent information derived from NASC-PET, which made the training more efficient.

Figure 11. The general protocol of our proposed domain knowledge integrated deep learning method. We decomposed the complex end-to-end generation from non-attenuation and non-scatter corrected images (NASC-PET) to corrected images (ASC-PET) into two components, anatomy-independent textures (relating to tracers and diseases) and anatomy-dependent correction, and regularized a 3D deep neural network to estimate this low-frequency anatomy-dependent correction information only. (Figure 5 in manuscript)

Following the suggestion, we state this in the Discussion section clearer: “We hypothesized that although NASC-PET images do not contain explicit information about photon attenuation and scattering, trained deep neural networks could predict low-frequency anatomy-dependent information which can be applied for correction. As shown in Figure 5, motivated by the physical principle of ACF and scatter contribution, our proposed Decomposition-based DL modified one step of the conventional 3D

method, i.e., we regularized the network to estimate low-spatial frequency anatomy-dependent information derived from NASC-PET, which is a voxel-wise ratio map. Detailed implementation would be described in the Method section. The comparison between Decomposition-based DL and conventional 3D DL can be viewed as an ablation study that demonstrated not only the advantages of our modification but also the possibility of incorporating domain knowledge into the design of artificial intelligence in a simple way. Meanwhile, since we decomposed the complex end-to-end generation from NASC-PET to ASC-PET into two components, anatomy-independent textures (relating to tracers and diseases) and anatomy-dependent correction, as a result, we posed an easier learning task for the model so that we could increase both the efficiency and robustness.”

3. *The proposed method of estimating a pixel-wise ration map can be interpreted as projecting the integrated attenuation of tissue around each voxel within each voxel. Increased robustness of the proposed method might therefore stem from the fact that only a rough estimate of surrounding tissue mass might be required to estimate this ration map, which might be an easier task to learn. Potentially, the described method might however deteriorate in unusually obese or cachectic patients. This should be investigated e.g. by plotting BMI-dependent quantification errors (the relatively inferior performance on the UI and GE scanners might be a hint in this direction as the respective populations have significantly different weight distributions compared to the training population).*

Response: Thanks for the Reviewer’s suggestions. It is indeed interesting to investigate the relationship between the BMI and prediction errors, we’ve added the results to the Supplementary material. As shown in Figure 12 and Table 1. According to the Pearson test, p values are larger than 0.05 on all datasets except Vision 600 (Bern)- 68Ga-DOTA-TOC, which contains only 8 subjects. The results imply that the performance of our model was not affected by the weight distribution of the subjects, as body mass index (BMI) was generally not associated with NRMSE. The different performance of the UI and GE scanners may be attributed to the variability of instrumentation and imaging protocols, such as geometric configuration, detector capability data correction, and system calibration.

Figure 12. Scatter plot of the relationship between body mass index and normalized root mean squared error in different scanners and tracers. (Supplementary FIGURE 5)

	Cross Scanner				Cross tracer			
Source	Vision 450 (SH)- FDG	Vision 600 (Bern)- FDG	UI uMI 780 (SH)- FDG	GE Discovery MI (SH)- FDG	Vision 450 (SH)- ⁶⁸ Ga-FAPI (SH)	Vision 450 (SH)- ⁶⁸ Ga-DOTA-TATE	Vision 600 (Bern)- ⁶⁸ Ga-DOTA-TOC	Vision 600 (Bern)- ¹⁸ F-PSMA
Pearson r	-0.2671	0.1836	0.03857	0.04052	0.7152	0.1845	0.8308	0.3618
P value (two-tailed)	0.0582	0.1531	0.7061	0.6830	0.0708	0.4784	0.0206	0.2478

Table 1. Pearson test relationship between body mass index (BMI) and normalized root mean squared error (NRMSE) in different scanners and tracers. BMI is not correlated with the NRMSE. (Supplementary Table S2)

4. *The approach proposed by the authors is somewhat similar to an alternative approach of estimating a pseudo-CT as an intermediate step. This alternative approach however actually adds prior knowledge as it represents the physical process of PET correction and allows for subsequent physics-based image reconstruction and*

correction. Therefore, this approach (pseudo-CT as an intermediate step) should be included as a baseline of comparison.

Response: Thanks for the kind reminder from Reviewer’s comments. We implemented the Pseudo-CT based approaches according to Fang etc.³⁰. As shown in Figure 13, the model was able to generate pseudo-CT from the input non-corrected PET images, but it is less accurate compared to 2D or 3D direct approaches since the trained model is trying to achieve cross-domain image synthesis. It imposed a much harder task for the neural network, considering the big difference between PET imaging and CT imaging. Also, the Pseudo-CT method requires further reconstruction steps, which may amplify the errors generated during Pseudo-CT generation. While our Decomposition-based DL or other conventional DL approaches only deal with the image generation within the same domain.

Figure 13. Comparison between our Decomposition-based method with a Pseudo-CT based approach on data scanned with Siemens Biograph Vision 450 ¹⁸F-FDG.

5. The approach for quantitative evaluation of algorithm accuracy is rather intransparent due to the following reasons:

- The most important metric for comparing methods of PET attenuation correction should be voxel-based activity measures (either as Bq/ml or SUV). Texture features / similarity metrics are of only secondary relevance to the task of PET attenuation/scatter correction.

- Quantitative PET values should be measured in different organs (e.g. using regions of interest) in order to provide a clear picture of the spatial distribution of quantification errors and potential anatomical variability.

- The errors should be provided as mean absolute errors (not as mean errors as these can cancel out) over each region of interest. The chosen metrics do not provide this

level of detail.

- Instead of providing scatter plots, it would be much more informative to provide voxel-wise maps of the absolute error for representative examples as this would again give insight into the spatial anatomical distribution of errors

- The authors do not describe how lesions were identified to measure related metrics: how many lesions were identified, how were they segmented, where were they located etc.

Response: Thanks for the Reviewer's comments. We agree that we should describe our evaluation method in more detail, so we added more information in Supplementary materials as shown in answer 12. To answer your questions:

- We evaluated our results (AI ASC-PET) to the CT ASC-PET using both voxel-wise global metrics (NRMSE, PSNR, and SSIM), where the unit of each voxel value in Bq/mL, among which NRMSE was considered the most relevant metric.

- Apart from the global physical metrics, we also evaluated local metrics of the volume of interest within targeted organs (kidney, liver, and heart). Here we converted the voxel value to SUV and evaluated SUV-related metrics, as well as radiomic features, including root mean squared error, median. Percentage error was used to evaluate the accuracy of each metric in reference to the CT ASC-PET.

- As for global metrics, normalized root mean squared error can be viewed similar to mean absolute error. As for local metrics, we did use mean absolute percentage error.

- We agree that plotting the difference image would make the pattern clearer, so we added Figure 4 into Supplementary material.

- We added this missing information on Page 8: "Furthermore, on six randomly selected subjects in each dataset, two certified nuclear medicine physicians (R.G. and C.M.) used ITK-SNAP to manually delineate three spherical VOIs within each target organ (liver, kidney, and heart)." We include this VOI delineation in the demo data as well, as shown in Figure 2.

Following the suggestion, we add 3 VOIs for each organ and updated the results in Figure 3. The updated results showed still that Decomposition-based DL outperformed the other two on all scanners and all tracers regarding local metrics, only that all groups had slightly higher error rates compared to previous results.

6. *Title: The title should be more specific. E.g. it is unclear what the term "trustworthy" means in this context. One suggestion might be: "Towards robust and generalizable machine learning-based CT-free PET attenuation correction"*

Response: Thanks for the suggestion. Following the suggestion, we modified the title to make it more specific as "Practice of domain knowledge towards robust and

generalizable deep learning-based CT-free PET attenuation and scatter correction”

7. *Abstract: The core idea of the method should be briefly described in the abstract. Similar to the rest of the manuscript it remains unclear what the described "decomposition" actually consists of.*

Response: Thanks for the Reviewer’s comments. We agree that the description of our proposed method should be stated clearer in Abstract. So we modified the sentence as following: “In contrast to conventional direct deep learning methods, we simplify the complex problem by a domain decomposition so that the learning of anatomy-dependent attenuation correction can be achieved robustly in a low-frequency domain while the original anatomy-independent high-frequency texture can be preserved during the processing. The effectiveness and robustness of our proposed approach was verified in tests of external imaging tracers on different scanners.”

8. *Introduction: The authors motivate their method in part by referring to total-body PET/CT scanners. However, it should be mentioned in this context that alternative approaches for CT-free PET correction are being developed for these scanners that rely on physical effects (e.g. https://jnm.snmjournals.org/content/62/supplement_1/1530)*

Response: Thank the Reviewer to point out alternative approaches for CT-free PET correction relying on physical effects. . We added these works in the introduction on Page 3 and addressed their limitations as follows: “Therefore, numerous research efforts have been devoted to developing methods for CT-free PET correction. Magnetic resonance (MR)-based approaches yielded satisfactory results in brain PET ²⁷. Alternatively, attenuation maps (μ -maps) can also be derived using the maximum likelihood estimation of activity and attenuation (MLAA) algorithm. MLAA can be further improved with the use of the additional time of flight (TOF) ^{17,18}. Due to the insufficient timing resolution of current clinical PET systems, the MLAA suffers from the crosstalk between the activity and attenuation distribution and high noise ¹⁹. Recent MLAA methods alleviate the low-frequency crosstalk problem ²⁰, but may over-smooth or over-estimate the bone structure ^{21,22}. TOF-MLAA-based approaches using an external source like rod ¹⁷ or lutetium oxyorthosilicate background transmission ²³ are under development.”

9. *In general the term AI (artificial intelligence) is rather vague. It seems preferable to use the term machine learning in the context of this work.*

Response: Thanks for the Reviewer’s comments. Based on this comment, we changed it more specifically to deep learning, in line with the title.

10. *Results: see above. The level of detail regarding used metrics and performed analysis is not sufficient for valid assessment of the proposed method.*

Response: Thanks for the Reviewer’s comments. We agree that our previous

description of our evaluation method was not clear enough to follow, following your suggestion, we've modified it in the manuscript on Page 9 as follows:

“To evaluate the quality of the DL ASC-PET images, we calculated the global physical metrics, to the CT ASC-PET, including voxel-wise NRMSE, PSNR, and SSIM. Furthermore, on six randomly selected subjects in each dataset, two certified nuclear medicine physicians (R.G. and C.M.) used ITK-SNAP to manually delineate three spherical VOIs within each target organ (liver, kidney, and heart). This was followed by SUV measurements as well as statistical analysis of clinical imaging parameters including SUVmean, SUVmax, total lesion metabolism (TLM), as well as the most relevant radiomics features. The performance of our proposed Decomposition-based DL method was further investigated through comparison with traditional direct DL approaches, differences between each group for NRMSE were assessed for statistical significance by means of the paired two-tailed t-test, with a statistically significant difference defined as $p < 0.05$ with Bonferroni correction for minimizing type I error. More information on the data preprocessing, network design, training procedure, and physical metrics is attached in the corresponding part of Supplementary material.”

We also described the used metrics in details in Supplementary material as shown in the “Modified Method section in Supplementary materials” attached after reference of this response letter.

11. Discussion: Mentioned limitations should be addressed. Also, alternative methods for increasing algorithm robustness should be discussed.

Response: Thanks for the Reviewer's comments. Based on the comment, we think deeply about our limitations. We discussed in the Discussion section as follows: “Meanwhile, the performance of our model is subject to NASC-PET, which implies that it will not perform well when dealing with extreme cases, such as patients with upside-down images, contaminated patients, patients with implants, or abnormally high activity in certain regions. Supplementary Figure 4 showed two extreme cases where our model overestimated the activity in the neck region, probably due to the abnormally high activity of NASC-PET images in the neck region. Data crafting, unsupervised domain adaptation may help with performance degradation on unseen images.”

The current limitation we mentioned here is that we “trained a model on a homogeneous dataset with only one scanner and one tracer, which is not optimal for the DL development”. We agree that an expanded dataset or trained on a mixed dataset would help either for the development of the model, or to fairly evaluate the model. Following your suggestion, we performed an experiment where we trained a model with a dataset mixed with Vision 450 (original training dataset) and an external scanner (GE DMI or UI 780), and we total amount of the subjects in the dataset was kept the same as the previous training set. As shown in Figure 14, the proposed

Decomposition-based method does benefit from this mixture training both quantitatively and qualitatively.

Figure 14. Results of Decomposition-based method with and without mixture training quantitatively and qualitatively.

Our study aims to perform a proof-of-concept, which tries to demonstrate the possibility that DL might be able to achieve this cross-scanner and cross-tracer application when trained only on a homogeneous dataset. The feasibility demonstrated in this study encourages future development by incorporating a much larger and heterogeneous dataset for the training and eventually may therefore support the design of more realistic studies in the future, by including a larger and heterogeneous dataset that is not limited by the center, scanner, tracer, disease, or body region.

12. *Methods: see above. The level of detail regarding the methods (specifically also the proposed algorithm itself) is not sufficient in order for the reader to understand and potentially reproduce this work. Also, evaluation metrics need to be described more precisely.*

Response: Thanks for the suggestion from the reviewer. We agree that including more

detailed information about our proposed method would improve the readability and reproducibility., so we modified the **Method** section and Supplementary material as shown in the previous answers. We also included more information about network design and training procedure in Supplementary material, shown in the “Modified Method section in Supplementary materials” attached after reference to this response letter.

13. *Methods: It is to be expected that the "AI-generated anatomy dependent correction map" does not capture background noise (as also seen in Figure 5). This should result in high image background noise levels in the AI-corrected PET due to existing noise in the non-corrected PET. Is this the case? If not why? This effect might also be relevant for low-count areas in the foreground. This should be further explained.*

Response: Thanks to the reviewers for their comments. To clear things out first, Following the comment, we looked into more details of the problem. For our method, we have limited effects on the noise levels. The training and prediction of anatomy-dependent correction map are in low resolution smoothed space and the results have usually low noise. After up-sampling of the generated ratio map and apply to NASC-PET images, the noise level of the result AC-PET should be similar to the NASC-PET images. We apologized for a mistake in our previous protocol diagram (Figure 5), where we did not illustrate the schematic presentation of the anatomy-dependent correction map (ADCM) in low-resolution space as expected. Now we updated the presentation of ADCM ratio map calculated after downsampling of the NASC PET and AC PET, where no such background noise can be seen. The noise-level in down-sampled low-resolution space is very low. Thanks to your comments, we’ve updated the protocol diagram (Figure 11) to avoid misunderstanding.

14. *Methods: The different scanners have different image characteristics (e.g. matrix size). How is this accounted for at test time? Are images resampled to similar spatial resolutions?*

Response: Many thanks to the Reviewer for pointing out this point of the potential convenience of our method in practice. After decomposition, the anatomy-dependent ratio map is learned and predicted at low resolution ($6.6 \times 6.6 \times 8$ mm/voxel), which is lower than almost all current scanners. Also, the input matrix size required for NASC PET ($112 \times 112 \times 112$) is lower than the matrix size of almost all current scanners. Therefore, our method is insensitive to the difference in matrix sizes or resolutions of different scanners. In practice, the NASC PET images are first down-sampled to the low-resolution and after AI-based prediction, the generated low-resolution anatomy-dependent ratio map is then up-scaled to the original resolution of the NASC PET images. For direction AI methods, we resample them to the same low-resolution for the training. We added these details and corresponding advantages in the Discussion as follows: “In addition, our design allows NASC-PET input at low resolution (6.6×6.6

× 8 mm/voxel), small matrix size (112 × 112 × 112), which are lower than almost all current scanners. Therefore, our method is insensitive to differences of matrix sizes or resolution and can be applied to different scanners.”

References

1. Fendler, W. P. *et al.* (68)Ga-PSMA PET/CT: Joint EANM and SNMMI procedure guideline for prostate cancer imaging: version 1.0. *Eur. J. Nucl. Med. Mol. Imaging* **44**, 1014–1024 (2017).
2. Adam, J. A. *et al.* EANM/SNMMI practice guideline for [18F] FDG PET/CT external beam radiotherapy treatment planning in uterine cervical cancer v1. 0. *Eur. J. Nucl. Med. Mol. Imaging* **48**, 1188–1199 (2021).
3. Paiva, F. G., do Carmo Santana, P. & Mourão, A. P. Evaluation of patient effective dose in a PET/CT test. *Appl. Radiat. Isot.* **145**, 137–141 (2019).
4. Prieto, E. *et al.* Ultra-low dose whole-body CT for attenuation correction in a dual tracer PET/CT protocol for multiple myeloma. *Phys. Medica* **84**, 1–9 (2021).
5. Federal Office for Public Health. Diagnostische Referenzwerte für nuklearmedizinische Untersuchungen. https://www.bag.admin.ch/dam/bag/de/dokumente/str/str-wegleitungen/drw/drw-nuk.pdf.download.pdf/Strahlenschutz_Wegleitung_DRW-Nuklearmedizin_DE.pdf.
6. Fahey, F. H., Treves, S. T. & Adelstein, S. J. Minimizing and communicating radiation risk in pediatric nuclear medicine. *J. Nucl. Med. Technol.* **40**, 13–24 (2012).
7. Robbins, E. Radiation risks from imaging studies in children with cancer. *Pediatr. Blood Cancer* **51**, 453–457 (2008).
8. Panagiotidis, E. *et al.* Comparison of the impact of 68Ga-DOTATATE and 18F-FDG PET/CT on clinical management in patients with neuroendocrine tumors. *J. Nucl. Med.* **58**, 91–96 (2017).
9. Pouliot, F. *et al.* The Triple-Tracer strategy against Metastatic Prostate cancer (3TMPO) study protocol. *BJU Int.* (2021).
10. Surasi, D. S. S., Lin, L., Ravizzini, G. & Wong, F. Supraclavicular and axillary lymphadenopathy induced by COVID-19 vaccination on 18F-fluorothalamic acid, 68Ga-DOTATATE, and 18F-Fluciclovine PET/CT. *Clin. Nucl. Med.* **47**, 195–196 (2022).
11. Jha, A. *et al.* Sporadic Primary Pheochromocytoma: A Prospective Intraindividual Comparison of Six Imaging Tests (CT, MRI, and PET/CT Using 68Ga-DOTATATE, FDG, 18F-FDOPA, and 18F-FDA). *Am. J. Roentgenol.* **218**,

342–350 (2022).

12. Zhou, Y., Baidoo, K. E. & Brechbiel, M. W. Mapping biological behaviors by application of longer-lived positron emitting radionuclides. *Adv. Drug Deliv. Rev.* **65**, 1098–1111 (2013).
13. Chomet, M. *et al.* Head-to-head comparison of DFO* and DFO chelators: Selection of the best candidate for clinical ⁸⁹Zr-immuno-PET. *Eur. J. Nucl. Med. Mol. Imaging* **48**, 694–707 (2021).
14. Alberts, I. *et al.* Clinical performance of long axial field of view PET/CT: a head-to-head intra-individual comparison of the Biograph Vision Quadra with the Biograph Vision PET/CT. *Eur. J. Nucl. Med. Mol. Imaging* **48**, 2395–2404 (2021).
15. Hwang, D. *et al.* Improving the accuracy of simultaneously reconstructed activity and attenuation maps using deep learning. *J. Nucl. Med.* **59**, 1624–1629 (2018).
16. Teimoorisichani, M. *et al.* A CT-less approach to quantitative PET imaging using the LSO intrinsic radiation for long-axial FOV PET scanners. *Med. Phys.* **49**, 309–323 (2022).
17. Panin, V. Y., Aykac, M. & Casey, M. E. Simultaneous reconstruction of emission activity and attenuation coefficient distribution from TOF data, acquired with external transmission source. *Phys. Med. Biol.* **58**, 3649 (2013).
18. Salomon, A., Goedicke, A., Schweizer, B., Aach, T. & Schulz, V. Simultaneous reconstruction of activity and attenuation for PET/MR. *IEEE Trans. Med. Imaging* **30**, 804–813 (2010).
19. Chun, S. Y., Kim, K. Y., Lee, J. S. & Fessler, J. A. Joint estimation of activity distribution and attenuation map for TOF-PET using alternating direction method of multiplier. in *2016 IEEE 13th International Symposium on Biomedical Imaging (ISBI)* 86–89 (IEEE, 2016).
20. Wang, G. & Qi, J. PET image reconstruction using kernel method. *IEEE Trans. Med. Imaging* **34**, 61–71 (2014).
21. Hwang, D., Kang, S. K., Kim, K. Y., Choi, H. & Lee, J. S. Comparison of deep learning-based emission-only attenuation correction methods for positron emission tomography. *Eur. J. Nucl. Med. Mol. Imaging* 1–10 (2021).
22. Li, S. & Wang, G. Modified kernel MLAA using autoencoder for PET-enabled dual-energy CT. *Philos. Trans. R. Soc. A* **379**, 20200204 (2021).
23. Teimoorisichani, M. *et al.* Using LSO background radiation for CT-less attenuation correction of PET data in long axial FOV PET scanners. (2021).
24. Ben Yedder, H., Shokoufi, M., Cardoen, B., Golnaraghi, F. & Hamarneh, G. Limited-angle diffuse optical tomography image reconstruction using deep

- learning. in *International conference on medical image computing and computer-assisted intervention* 66–74 (Springer, 2019).
25. Yang, G. *et al.* DAGAN: deep de-aliasing generative adversarial networks for fast compressed sensing MRI reconstruction. *IEEE Trans. Med. Imaging* **37**, 1310–1321 (2017).
 26. Kamnitsas, K. *et al.* Efficient multi-scale 3D CNN with fully connected CRF for accurate brain lesion segmentation. *Med. Image Anal.* **36**, 61–78 (2017).
 27. Ladefoged, C. N. *et al.* A multi-centre evaluation of eleven clinically feasible brain PET/MRI attenuation correction techniques using a large cohort of patients. *Neuroimage* **147**, 346–359 (2017).
 28. Lau, W. L., Liang, C., Liu, H., Singh, K. & Mukherjee, J. Development of zirconium-89 PET for in vivo imaging of alpha-klotho. *Am. J. Nucl. Med. Mol. Imaging* **10**, 95 (2020).
 29. Dehdashti, F. *et al.* Evaluation of [⁸⁹Zr] trastuzumab-PET/CT in differentiating HER2-positive from HER2-negative breast cancer. *Breast Cancer Res. Treat.* **169**, 523–530 (2018).
 30. Liu, F. *et al.* A deep learning approach for 18 F-FDG PET attenuation correction. *EJNMMI Phys.* **5**, 1–15 (2018).

Modified **Method** section in Manuscript:

Decomposition-based DL

We decomposed the complex end-to-end generation from NASC-PET to ASC-PET into two components, anatomy-independent textures (relating to tracers and diseases) and anatomy-dependent correction, and regularized a 3D deep neural network to estimate this low-frequency anatomy-dependent correction information only.

As shown in Figure 5, we first calculated the anatomy-dependent correction map (ADCM) based on NASC-PET and ASC-PET according to the following equations:

If $I^{\text{NASC-PET}}[x, y, z] > \varepsilon$ then

$$I^{\text{ADCM}}[x, y, z] = \frac{I^{\text{ASC-PET}}[x, y, z]}{I^{\text{NASC-PET}}[x, y, z]}$$

else $I^{\text{ADCM}}[x, y, z] = I^{\text{ASC-PET}}[x, y, z]$

where we set ε to be 1. In order to preserve more spatial information, which is most essential for the task of attenuation and scatter correction, we downsampled the NASC-PET and ADCM to size of $112 \times 112 \times 112$. A semi-supervised 3D conditional generative adversarial network (c-GAN) was employed, which consists of a generator network (G) to synthesize the ADCM from NASC-PET, and a discriminator (D) to distinguish between the synthesized ADCM and the real inputs, the objective function is defined as:

$$\begin{aligned} \min_G \max_D V(D, G) + \lambda V(G) \\ = \sum_{i=1}^n \log(D_{\theta_D}(I^{\text{NASC-PET}}, I^{\text{ADCM}})) \\ + \log(1 - D_{\theta_D}(I^{\text{NASC-PET}}, G_{\theta_G}(I^{\text{NASC-PET}}))) + \lambda |G_{\theta_G}(I^{\text{NASC-PET}}) - I^{\text{ADCM}}|_2 \end{aligned}$$

where λ is a weighting of loss, which was set to $1e+4$ based on experiments. The model was trained on 470 3D images of size $112 \times 112 \times 112$ and validated on 51 reserved subjects (Table 1).

In the testing stage, given a NASC-PET, the trained generator networks G was used to predict the DL-generated ADCM ($I^{\text{DL-ADCM}}$), and applied to NASC-PET to obtain DL ASC-PET according to the following equations:

$$I^{\text{DL ASC-PET}}[x, v, z] = I^{\text{NASC-PET}}[x, v, z] * I^{\text{DL-ADCM}}[x, v, z]$$

To evaluate the quality of the DL ASC-PET images, we calculated the global physical metrics, to the CT ASC-PET, including voxel-wise NRMSE, PSNR and SSIM. Furthermore, on six randomly selected subjects in each dataset, two certified nuclear medicine physicians (R.G. and C.M.) used ITK-SNAP to manually delineate three spherical VOIs within each target organ (liver, kidney, and heart). This was followed by SUV measurements as well as statistical analysis of clinical imaging parameters including SUV_{mean} , SUV_{max} , total lesion metabolism (TLM), as well as the most relevant radiomics features. The performance of our proposed Decomposition-based DL method was further

investigated through comparison with traditional direct DL approaches, differences between each group for NRMSE were assessed for statistical significance by means of the paired two-tailed t -test, with a statistically significant difference defined as $p < 0.05$ with Bonferroni correction for minimizing type I error. More information on the data preprocessing, network design, training procedure and physical metrics is attached in corresponding part of Supplementary material.

Modified **Method** section in Supplementary materials:

Decomposition-based DL

As shown in Figure 5, we first calculated the anatomy-dependent correction map (ADCM) based on non-attenuation and non-scatter corrected image ($I^{\text{NASC-PET}}$) and corrected PET ($I^{\text{ASC-PET}}$), which is a voxelwise ratio map (I^{ADCM}).

In order to preserve more spatial information, which is most essential for the task of attenuation and scatter correction, we zero-padded and downsampled the NASC-PET and ADCM to size of $112 \times 112 \times 112$ from $440 \times 440 \times 448$. Min-max normalization was applied to the entire dataset. To be noted that, when testing on external scanners, images are first resampled to the same voxel spacing as training data ($1.65 \times 1.65 \times 2$ mm), and then zero-padded and resized to $112 \times 112 \times 112$.

Deep Neural Network design

The goal of the generator is to be able to approximate the corresponding ratio map for a given NASC-PET images, while the discriminator aims to distinguish between the synthesized ratio map and the real input.

Generator network: The goal of the generator is to be able to approximate the corresponding I^{ADCM} for a given $I^{\text{NASC-PET}}$. We built layers with multiple Convolution-Relu components. Specifically, the entire network constitutes 15 convolutional layers. In the encoder part which includes the first 8 convolutional layers. The number of feature maps increases from 64 in the 1st layer to 512 in the 8th layer,

once every two layers, we use $3 \times 3 \times 3$ filters and a stride of 2 every two layers. In the decoder part, we perform up-sampling with a factor of 2. Using the skip connections, the feature maps from the encoder part are copied and concatenated with the feature maps of the decoder part.

Discriminator network: The discriminator network aims to distinguish between the I^{s-ADCM} generated by the generator model, and the real input I^{ADCM} . The discriminator takes either a ADCM or a synthesized one as input and determines whether the input is real or not. The architecture of the discriminator network contains eight convolution blocks and a fully connected block at the end. The last sigmoid activation output a probability to determine whether the input is real or synthetic. The discriminator was built with multiple convolution-batch normalizaion-Leaky Relu components. 0.2 negative slope was set for the leaky ReLu and 0.8 momentum for the batch normalization.

$$\begin{aligned} \min_G \max_D V(D, G) &= \sum_{i=1}^n \log \left(D_{\theta_D} (I^{NASC-PET}, I^{ADCM}) \right) \\ &+ \log \left(1 - D_{\theta_D} \left(I^{NASC-PET}, G_{\theta_G} (I^{NASC-PET}) \right) \right) \end{aligned}$$

In contrast to conditioning the generation of images on random noise drawn from specific distribution, this objective function takes an input of NASC-PET image. Furthermore, the adversarial loss of our model also included voxel-wise content loss alongside image-wise loss, to ensure spatial alignment of the generated ratio map with the ground truth:

$$L_{\text{content loss}} = \sum_{i=1}^n |I^{\text{DL-ADCM}} - I^{\text{ADCM}}|_2$$

$$\text{where } I^{\text{DL-ADCM}} = G_{\theta_G} (I^{\text{NASC-PET}})$$

Therefore, the overall objective function was defined as:

$$\begin{aligned}
& \min_G \max_D V(D, G) + \lambda V(G) \\
& = \sum_{i=1}^n \log \left(D_{\theta_D} (I^{\text{NASC-PET}}, I^{\text{ADCM}}) \right) \\
& \quad + \log \left(1 - D_{\theta_D} \left(I^{\text{NASC-PET}}, G_{\theta_G} (I^{\text{NASC-PET}}) \right) \right) \\
& \quad + \lambda \|G_{\theta_G} (I^{\text{NASC-PET}}) - I^{\text{ADCM}}\|_2
\end{aligned}$$

Training details: We employed the Adam solver with a batch size of 1 and a learning rate of 0.0002. In order to facilitate efficient access to this large number of images during training, the dataset was organized into a single data object in HDF5 (Hierarchical Data Format 5). All of our experiments were implemented in TensorFlow and trained on our NVIDIA GeForce GTX 1080 Ti graphic cards. The training process ran for 100 epochs, and the weight of the content loss λ was set to $1e+4$ based on experiments.

Evaluation based on Physical Metrics, Clinical and Radiomics features

Physical Metrics

To evaluate the quality of the DL ASC-PET images, we calculated and compared the following metrics: 1. Normalized root mean squared error (NRMSE); 2. Peak signal-to-noise ratio (PSNR); 3. Structural similarity index measurement (SSIM). The NRMSE is defined as:

$$NRMSE = \frac{\sqrt{\frac{\sum_{i=1}^n (y_{true} - y_{pred})^2}{n}}}{\max(y_{true}) - \min(y_{pred})}$$

where y_{true} is the CT ASC-PET and y_{pred} is the DL ASC-PET image, and it measures the overall pixel-wise intensity deviation between these two. The PSNR is defined as:

$$PSNR = 10 \log_{10} \left(\frac{VR^2}{\|y_{true} - y_{pred}\|_2^2} \right)$$

where V is the total amounts of voxels and R represents the range of the intensity of the CT ASCT-PET image, and $\|y_{true} - y_{pred}\|_2^2$ computes the mean squared error between it and the DL ASC-PET image. The pixel-wise quantities are easily calculated and compared and have straightforward interpretations. However, they do not correspond well with the sort of errors that humans perceive, particularly blurring and smearing artifacts, and images with identical NRMSE values may appear substantially different. Additional measures that more accurately reflect perceived image quality are therefore desirable.

$$SSIM(x, y) = \frac{(2\mu_x\mu_y + C_1)(2\sigma_{xy} + C_2)}{(\mu_x^2 + \mu_y^2 + C_1)(\sigma_x^2 + \sigma_y^2 + C_2)}$$

where μ_x , μ_y are the averages of images CT ASC-PET and DL ASC-PET, and σ_x , σ_y are their standard deviations, respectively. C_1 and C_2 are two positive constants to avoid a null denominator. Theoretically, image with lower NRMSE, higher PSNR and SSIM closer to 1 represent higher synthesis quality.

Clinical and Radiomics features

Spherical VOIs were manually delineated within targeted organs (liver, kidney and heart) by two board certified nuclear medicine physicians (R.G. and C.M.) using ITK-SNAP. This was followed by standardized uptake value (SUV) discretization as well as statistical analysis. Clinical parameters and radiomics features were both included for the analysis, and we selected the mostly applied features based on references.

Clinical features including SUV_{mean} , SUV_{max} , total lesion metabolism (TLM), and radiomics features including root mean Squared, 90Percentile, median, joint average from gray-level co-occurrence matrix (GLCM), high gray level run emphasis from gray level Run length matrix (GLRLM), and zone percentage from gray level size zone (GLSZM). The accuracy of the clinical imaging parameters and radiomics features of the lesions within targeted organs was calculated in reference to the CT ASC-PET using mean absolute percentage error).

Reviewers' Comments:

Reviewer #1:

None

Reviewer #2:

Remarks to the Author:

Based on the previous reviews' comments, authors made detailed supplements to the core ideas, methods, and experimental details, which contributed to the readability and reproducibility of the manuscript.

The deep learning-based method proposed by the author realizes CT-free PET imaging. The low-frequency anatomical structure is separated from the high-frequency texture part by domain decomposition, so that the deep learning method only trains the low-frequency anatomical structure, thus obtaining the anatomy-dependent correction map (ADCM). The test results of external experiments are significantly better than traditional deep learning methods, whether on external scanners or external tracers, verifying the generalization and robustness of the proposed method.

However, there are still several problems in the manuscript that need to be further revised and confirmed.

1. Abstract, from the experimental results, the diversity of tracer drugs has a greater impact on image reconstruction than scanners. In response to this problem, the author may discuss how the following work may alleviate the influence of tracers on image reconstruction.

2. Results, for traditional deep learning methods, the results from 2D and 3D DL methods in Figure 1/2 seem to be similar, but in the results shown in Figure 3/4, most of them point to 3D methods being better than 2D methods, which appears to be inconsistent. At the same time, although limited by the computing resources of the laboratory, there exist public servers (including better computing resources) that can be rented, and the author can have a better training environment to achieve better imaging results of the 3D network.

3. Methods, the proposed method is to divide the image after attenuation scatter correction with the image before correction to get the ADCM, then the network gets the ADCM map from the NASC-PET training, and finally multiply it back. This is actually a bit similar to the traditional end-to-end method, except that the input and output are changed, which is a clever way to separate the anatomy and texture parts. And the effect of the proposed method is significant. In addition, if you subtract the two images ASC-PET and NASC-PET, you can also get the difference map of the two, and then add it back after the network is trained. What will be the effect?

4. SUPPLEMENTARY MATERIAL, line 55, batchsize = 1, is it better to use instance normalization?

Reviewer #3:

Remarks to the Author:

As part of the revision process the authors provided additional information and performed additional experiments and analyses that contribute to a better understanding of the proposed method and its value.

Most importantly, the authors now provide a precise definition of the method in mathematical terms. Based on this definition the core element of the proposed method becomes clear: The main idea is to estimate a ratio map between corrected and uncorrected PET from the uncorrected PET instead of directly estimating the corrected PET from the uncorrected PET. It is conceivable - as the authors now explain - that this task might be easier to learn for a neural network and thus increase robustness.

(1) However - now that the approach is clear - I do not see advantages of this approach compared to estimating an attenuation map (instead of the ADCM ratio map). The attenuation map would have only advantages: (i) its values depend only on regional tissue properties (attenuation) and should thus be easier to learn compared to the ratio map, where the values also depend on global image properties (location of tissue), (ii) local errors in an estimated attenuation map would not directly propagate to the corrected PET image due to the subsequent reconstruction step, (iii) it is easier for a human to identify errors or artifacts in an attenuation map compared to a ratio map. To provide a methodological perspective: An attenuation map is also a ratio map, but in the

original acquisition space (sinogram space) resulting in these advantages.

Thus, my main request would be that the authors compare their approach to the relevant reference standard of estimating a pseudo attenuation map (this is not the same as estimating a pseudo CT)

Beyond this main point I have the following important comments:

(2) In order to make their method work, the author need to introduce an artificial threshold ϵ (which would not be necessary when estimating a pseudo attenuation map, see 1). They set this parameter to 1. This means that the authors actually estimate a mixed image between a corrected PET (for values below 1) and a ratio map (for values above 1). However, there are relevant tissue types in the human body (e.g. lung tissue) that usually have PET values below 1. The proposed method would introduce significant errors in these regions, which can be clinically relevant.

(3) I have problems in resolving inconsistencies in the results, which require further clarification. For example:

(i) Figure 9: I would expect the ADCM to show a gradient with increasing values towards the center of the body in order to make up for the increased attenuation in central areas (which can be see e.g. in the non-corrected PET in Figure 6, left). However, this does not seem to be the case (e.g. the liver is fairly homogeneous). Also - being a ratio map - the ADCM should mainly reflect the attenuation and the position (superficial or central) of an anatomic region and not its PET signal intensity. However, Figure 9 clearly shows that e.g. the bladder has high values in the ADCM reflecting the PET signal (also e.g. kidney and liver). Figure 9 should show the non-corrected and the corrected PET in addition. Also, the authors should always depict the ADCM, which is the central aspect of this method in all examples.

(ii) also in Figure 9: It is not credible that the ADCM shown in Figure 9 is estimated from a PET image. The depicted ADCM shows details of e.g. bone structures and lung vessels, which are not seen on PET data. This is obviously a ground-truth ADCM of some kind (see (i)). Why do the authors not show estimated ADCM maps (in non of the examples)? These should be part of every image example - they are the central aspect of this work.

(iii) Figure 13: The CT and pseudo CT do not show the same patient as the PET data. Also, the quality of the estimated pseudo CT is not state-of-the art. Most importantly, the authors should estimate a pseudo attenuation map instead of a pseudo CT (see 1)).

(iv) Figure 12 (bottom) shows a positive correlation between BMI and error: This supports the hypothesis that estimating the ADCM requires global information, which is a limitation of the method; estimating pseudo attenuation map would not have this limitation (see 1)

(4) Overall, I want to suggest the use of clear terminology that reflects what is actually shown. E.g. instead of an "anatomy dependent correction map" the term ratio map would be more intuitive. Also, the authors use terms like "low-frequency components" which are (in a strict sense) not related to the presented method. Importantly, I still cannot identify where domain knowledge is injected in this approach. Domain knowledge would be injected if e.g. a pseudo attenuation map is estimated because then the physical knowledge would be used for reconstruction. It seems to me that the proposed method in comparison rather neglects this domain knowledge resulting in the above-mentioned issues (1)

Response to Review Comments on NCOMMS-22-03233A

Response: We would like to thank again the Editors and Reviewers for their time and effort on our manuscript and for allowing us to revise this manuscript. Their valuable and constructive comments helped us to improve this study. Following the comments, we have substantially revised the manuscript and have addressed the comments carefully. In particular, we:

1. Included more datasets (^{68}Ga -DOTA-TOC from Vision 450-SH) for cross-scanner and cross-tracer evaluation.
2. Implemented μ -map based attenuation correction methods and compared the results to our proposed Decomposition-based deep learning (DL) method.
3. Added more analysis and explanations related to the mechanisms of the proposed method with the visualizations and investigations of the anatomy-dependent correction map (ADCM).

We hope that the revision will allow the manuscript to be accepted for publication.

Comments from Reviewers:

Reviewer #2:

Based on the previous reviews' comments, authors made detailed supplements to the core ideas, methods, and experimental details, which contributed to the readability and reproducibility of the manuscript.

The deep learning-based method proposed by the author realizes CT-free PET imaging. The low-frequency anatomical structure is separated from the high-frequency texture part by domain decomposition, so that the deep learning method only trains the low-frequency anatomical structure, thus obtaining the anatomy-dependent correction map (ADCM). The test results of external experiments are significantly better than traditional deep learning methods, whether on external scanners or external tracers, verifying the generalization and robustness of the proposed method.

However, there are still several problems in the manuscript that need to be further revised and confirmed.

Response: We appreciate the Reviewer for the generally positive assessment and encouragement of our study. We added the missing details in this revision according to the constructive comments.

1. Abstract, from the experimental results, the diversity of tracer drugs has a greater impact on image reconstruction than scanners. In response to this problem, the author may discuss how the following work may alleviate the influence of tracers on image reconstruction.

Response: We thank the Reviewer to guide us to think deep about our work. We agree

that the influence of diverse tracers is greater than scanners in application and did more test in this direction.

To alleviate the influence of tracers, we're planning to include a larger and heterogeneous dataset trained with a variety of tracers. The development is ongoing and will be published in the following paper. Meanwhile, with the current available dataset, we've conducted experiments to prove the feasibility of hybrid training. We trained two different Decomposition-based deep learning models: 1. Trained with 50 subjects of Vision 450 ^{18}F -FDG; 2. Trained with 50 subjects including (26 ^{18}F -FDG + 7 ^{68}Ga -FAPI + 17 ^{68}Ga -DOTA-TATE). Both models were tested with a newly collected dataset, including four subjects with ^{68}Ga -DOTA-TOC, scanned with Biograph Vision 450 in Shanghai. As shown in Figure 1, clearly the model trained with heterogeneous dataset outperformed the other one, suggesting that a dataset including various tracers would further improve the performance of our proposed model.

Figure 1. Quantitative comparison between two different Decomposition-based DL models when tested on ^{68}Ga -DOTA-TOC. ^{18}F -FDG only: model trained with homogeneous ^{18}F -FDG dataset. Hybrid: model trained with heterogeneous dataset (26 ^{18}F -FDG + 7 ^{68}Ga -FAPI + 17 ^{68}Ga -DOTA-TATE).

Additionally, we would like to point out that, the worse results on cross-tracer datasets compared to cross-scanner datasets partially resulted from the stacked influence (different tracers, different scanner). With the newly collected dataset, we were able to correct this effect, by comparing the performance of our model on three datasets: ^{68}Ga -DOTA-TOC (SH, Vision 450), ^{68}Ga -DOTA-TATE (SH, Vision 450) and ^{68}Ga -DOTA-TOC (Bern, Vision 600). As shown in Figure 2, our model achieved similar level of NRMSE on the two datasets from Vision 450 (SH), while outperformed the other one from Vision 600 (Bern).

Test on external tracers

Figure 2. Results of imaging from DOTA-TATE (Vision 450-SH), DOTA-TOC (Vision 450-SH) and DOTA-TOC (Vision 600-Bern).

2. Results, for traditional deep learning methods, the results from 2D and 3D DL methods in Figure 1/2 seem to be similar, but in the results shown in Figure 3/4, most of them point to 3D methods being better than 2D methods, which appears to be

inconsistent. At the same time, although limited by the computing resources of the laboratory, there exist public servers (including better computing resources) that can be rented, and the author can have a better training environment to achieve better imaging results of the 3D network.

Response: Thanks to the Reviewer for the insightful comments. Admittedly, Figures 1/2 shows that the 2D model achieved better average physical metrics compared to the 3D model, while the visual readings (Figure 3) or scattering histograms (Figure 4) shows that the 3D model is slightly better. In fact, as shown in Table 1, the physical metrics of the exemplary test results in Figure 3/4 shows that 2D model performed better. The reason behind this might be related to the fact that the 2D method, while preserving better detailed texture information, tended to underestimate the activities in organs and overestimate the neck and leg parts, due to the lack of spatial information during training. The 3D method did better in recovering the activity in organs but rendered blurry and over-smoothed images. Thus, when showed in the lower-resolution figures in the manuscript, results of 3D method might seem to be better. And the blurred and averaged results of the 3D method may also perform better in terms of scattering histograms.

NRMSE (%)		Conventional 2D	Conventional 3D	Decomposition DL
Cross Scanner	Vision 450 (SH)- FDG	0.14	0.21	0.09
	Vision 600 (Bern)- FDG	0.17	0.18	0.07
	UI uMI 780 (SH)- FDG	0.16	0.16	0.12
	GE Discovery MI (SH)- FDG	0.20	0.22	0.13
Cross Tracer	Vision 450 (SH)- 68Ga-FAPI	0.12	0.17	0.09
	Vision 450 (SH)- 68Ga-DOTA-TATE	0.12	0.12	0.10
	Vision 600 (Bern)- 68Ga-DOTA-TOC	0.24	0.25	0.12
	Vision 600 (Bern)- 18F-PSMA	0.86	0.84	0.29

Table 1. Normalized root mean squared error (NRMSE) results of exemplary test results of cross scanner and cross tracer tests. .

As for the training procedure, it is indeed one of our limitations that we did not train a 3D method. We face such limitations due to the following reasons: the model we designed with downsampled 3D image requires at least 7 GB of GPU memory. Training a 3D network with the entire 3D volume requires at least 448 GB of memory. Although some public servers can provide much larger GPU memory, our problem is that the

current data management does not allow training on public servers, which is the same situation for many institutions like ours. Exporting or uploading data in cloud computing servers may not be allowed.

Meanwhile, the design of downsampling is consistent with CT-based attenuation correction method. CT images typically require downsampling and smoothing to obtain an attenuation map^{1,2}. Our proposed Decomposition-based DL method works better with downsampled images, and it retains many other advantages by downsampling: 1. training and prediction of anatomy-dependent correction map (ADCM) in low resolution smoothed space helped to preserve at low background noise level. After up-sampling of the generated low-resolution ratio map and apply to NASC-PET images, the background noise level of the result AC-PET should be similar to the NASC-PET images; 2. our method is insensitive to the differences in matrix sizes or resolutions across scanners thanks to downsampling. Since the anatomy-dependent ratio maps were learned and predicted at low resolution ($6.6 \times 6.6 \times 8$ mm/voxel), which is lower than almost all current scanners.

3. *Methods, the proposed method is to divide the image after attenuation scatter correction with the image before correction to get the ADCM, then the network gets the ADCM map from the NASC-PET training, and finally multiply it back. This is actually a bit similar to the traditional end-to-end method, except that the input and output are changed, which is a clever way to separate the anatomy and texture parts. And the effect of the proposed method is significant. In addition, if you subtract the two images ASC-PET and NASC-PET, you can also get the difference map of the two, and then add it back after the network is trained. What will be the effect?*

Response: Thanks for the Reviewer's comments. The design of ratio-based method was motivated by the principle of attenuation and scattering correction. The iterative reconstruction algorithms can be considered as inverse estimation of the original image given the system matrix and measurements³: $e_m = e * att$, where e_m is the emission projection, att is the attenuation factor and e is the ideal (uncorrected) projection.

Following the Reviewer's comment, we implemented the subtraction method you mentioned. Figure 3 shows that the subtraction method (referred to here as residual 3D) performs slightly better than the conventional 3D DL direct method, but worse than the decomposition-based DL method. As discussed in the previous answer, the design of downsampling may favor a ratio-based approach over a subtraction approach.

Figure 3. Comparison between different DL approaches, including newly implemented Residual 3D DL method.

4. *SUPPLEMENTARY MATERIAL*, line 55, $batchsize = 1$, is it better to use instance normalization?

Response: Thanks for the Reviewer’s suggestion. We tested instance normalization (IN) during the development. Here we showed the comparison of two normalization method in Figure 4, the two networks performed similar on Vision 450 dataset (validation dataset), but the one with IN performed worse on cross-scanner datasets. The reason may be related to the fact that batch normalization adds additional noise to the training⁴, which may add additional robustness to the model.

Test on external scanners

Figure 4. Comparison between models with instance normalization and batch normalization.

Reviewer #3:

As part of the revision process the authors provided additional information and performed additional experiments and analyses that contribute to a better understanding of the proposed method and its value.

Most importantly, the authors now provide a precise definition of the method in mathematical terms. Based on this definition the core element of the proposed method becomes clear: The main idea is to estimate a ratio map between corrected and uncorrected PET from the uncorrected PET instead of directly estimating the corrected

PET from the uncorrected PET. It is conceivable - as the authors now explain - that this task might be easier to learn for a neural network and thus increase robustness.

Response: We appreciate the Reviewer for the recognition of our improvements.

1. However - now that the approach is clear - I do not see advantages of this approach compared to estimating an attenuation map (instead of the ADCM ratio map). The attenuation map would have only advantages: (i) its values depend only on regional tissue properties (attenuation) and should thus be easier to learn compared to the ratio map, where the values also depend on global image properties (location of tissue), (ii) local errors in an estimated attenuation map would not directly propagate to the corrected PET image due to the subsequent reconstruction step, (iii) it is easier for a human to identify errors or artifacts in an attenuation map compared to a ratio map. To provide a methodological perspective: An attenuation map is also a ratio map, but in the original acquisition space (sinogram space) resulting in these advantages.

Thus, my main request would be that the authors compare their approach to the relevant reference standard of estimating a pseudo attenuation map (this is not the same as estimating a pseudo CT)

Response: Thanks for the Reviewer to point out deep insight into our method and its difference to attenuation map. Following the suggestion, we tested the pseudo attenuation map method. We trained a 2D and a 3D network to synthesize pseudo attenuation map from NAC PET images directly, rather than pseudo CT, and subsequently used for PET reconstruction. To fairly compare to our method, we used the same network structure as ours. As shown in Figure 5, the model was able to generate pseudo μ -map from the input non-corrected PET images. However, the preliminary results of 5 test data (Figure 6) show that reconstructed PET images are less accurate compared to our Decomposition based method. An extensive comparison of the two methods is ongoing and will be published in the following paper.

We hope that the Reviewer might allow us to share some different opinions.

- (i) We agree that the learning of an ADCM ratio map needs to consider the global anatomy. However, the learning of attenuation maps from PET imaging using AI also needs global anatomical information. There is no direct quantitative mapping between PET uptake and attenuation values. The more anatomical information involved, the better attenuation map we may generate. That is also the principle why atlas-based method often shows better accuracy in attenuation correction⁵. Even for the segmentation-based method, the key point is to recognize the anatomical structure during the segmentation, where more global information is helpful.
- (ii) We agree that the errors in the ADCM ratio map can be propagated to the final images. However, the errors in attenuation map can be also propagated to the

final images⁶. The reconstruction procedure does not correct the errors. Indeed, small errors in attenuation map may be amplified more than ADCM ratio map⁷. In contrast, the errors in ADCM ratio map are proportional in the final images. The pseudo attenuation map method needs more operations (forward projection and backward correction) to achieve the corrections, which complicates the overall task. While our Decomposition-based DL method, the correction route is much shorter. The easier task may lead to better robustness when applied to external datasets, which is the essential advantage of our method compared to traditional direct DL method.

- (iii) We agree that humans get more used to attenuation map than ADCM ratio map. However, physicians or physicists still needs to be trained to screen the errors in attenuation map, which is not the same as CT images. Similarly, experience about quality control of ADCM ratio map may be gained with specific training.
- (iv) In contrast to attenuation map, the ADCM ratio map does not need the special vendor reconstruction software. According to our experience with Siemens, only limited collaborative centers can get the special software E7 tools to import the synthesized attenuation map in the reconstruction. It requires raw data from the scanners and dedicated storage space, a technician or physicist familiar with the reconstruction tool, and takes longer time to perform. While the ADCM ratio map can be directly applied on imaging space, which can be widely available in almost all centers.

Figure 5. Comparison between our Decomposition-based method with pseudo μ -map based methods.

Figure 6. Quantitative accuracy comparison of Decomposition-based method with a pseudo μ -map based methods.

2. Beyond this main point I have the following important comments: (2) In order to make their method work, the author need to introduce an artificial threshold ϵ (which would not be necessary when estimating a pseudo attenuation map, see 1). They set this parameter to 1. This means that the authors actually estimate a mixed image between a corrected PET (for values below 1) and a ration map (for values above 1). However, there are relevant tissue types in the human body (e.g. lung tissue) that usually have PET values below 1. The proposed method would introduce significant errors in these regions, which can be clinically relevant.

Response: Thanks for the Reviewer's insights. The artificial threshold ϵ was mainly used to avoid zero dividing issue. We apologized for missing a detail of our method. In fact, when applying the DL-generated ADCM ($I^{\text{DL-ADCM}}$) to NASC-PET, we used the same ϵ , to make sure the obtained DL ASC-PET pick up the activity from NASC-PET if below the threshold:

If $I^{\text{NASC-PET}}[x, y, z] > \epsilon$ then

$$I^{\text{DL ASC-PET}}[x, y, z] = I^{\text{NASC-PET}}[x, y, z] * I^{\text{DL-ADCM}}[x, y, z]$$

else $I^{\text{DL ASC-PET}}[x, y, z] = I^{\text{NASC-PET}}[x, y, z]$

During the development, we tested four different ways to calculate the ratio map, Figure 7 shows that the model achieved the best accuracy when ϵ was set to 1. Also, we tested four different ways how to apply the generated ratio maps to the NAC images. The results showed no significant differences. Therefore, we set ϵ to 1 for both calculating and applying the ratio map.

Figure 7. Results of different manners of calculating and applying ratio map.

3. I have problems in resolving inconsistencies in the results, which require further clarification. For example:

(i) Figure 9: I would expect the ADCM to show a gradient with increasing values towards the center of the body in order to make up for the increased attenuation in central areas (which can be seen e.g. in the non-corrected PET in Figure 6, left). However, this does not seem to be the case (e.g. the liver is fairly homogeneous). Also - being a ratio map - the ADCM should mainly reflect the attenuation and the position (superficial or central) of an anatomic region and not its PET signal intensity. However, Figure 9 clearly shows that e.g. the bladder has high values in the ADCM reflecting the PET signal (also e.g. kidney and liver). Figure 9 should show the non-corrected and the corrected PET in addition. Also, the authors should always depict the ADCM, which is the central aspect of this method in all examples.

(ii) also in Figure 9: It is not credible that the ADCM shown in Figure 9 is estimated from a PET image. The depicted ADCM shows details of e.g. bone structures and lung vessels, which are not seen on PET data. This is obviously a ground-truth ADCM of some kind (see (i)). Why do the authors not show estimated ADCM maps (in none of the examples)? These should be part of every image example - they are the central aspect of this work.

(iii) Figure 13: The CT and pseudo CT do not show the same patient as the PET data. Also, the quality of the estimated pseudo CT is not state-of-the art. Most importantly, the authors should estimate a pseudo attenuation map instead of a pseudo CT (see 1).

(iv) Figure 12 (bottom) shows a positive correlation between BMI and error: This supports the hypothesis that estimating the ADCM requires global information, which is a limitation of the method; estimating pseudo attenuation map would not have this limitation (see 1)

Response: Thanks for the Reviewer's insightful comments, which helped us to think deeper about the principle of our ADCM maps.

(i) We fully agree that ADCM should reflect a gradient, i.e. an increase in value towards the center of the body. In the previous figure, we showed only one coronal slice of the subject and this gradient was not so obvious due to the contrast of the picture. We plotted the coronal maximum intensity projection (MIP), as shown in Figure 8, and the MIP shows a clear increase in value toward the center of the body.

Figure 8. Example of the anatomy-dependent correction map (ADCM) and CT-based μ -map with Siemens Biograph Vision Quadra ^{18}F -FDG.

Meanwhile, the ADCM ratio map not only reflects the attenuation, but is also used to compensate for scattering. The scattering is not only dependent on the body tissue density, but the scatter correction is based on the radioactivity distribution in the body ^{8,9}. As shown in Figure 9, we have performed the reconstruction with/without scattering or attenuation correction and generated the corresponding ratio maps. It shows that the AC ratio map does not depend on the intensity, while the scattering ratio map relatively depends on the intensity. Furthermore, areas with denser regions (liver areas behind

bone structures of the rib cage) may suffer from scatter overcorrection which is a known problem ¹⁰. This overcorrection is mainly visible around the bladder where a high concentration of the radiopharmaceutical leads to halo artifacts (photopenic areas) ¹¹. These observations are consistent along a multitude of patients which further enhances the robustness of our results.

Figure 9. Example of PET with/without scatter or attenuation correction, as well as corresponding ratio map with Siemens Biograph Vision 600 ¹⁸F-FDG. Reconstructed PET image without attenuation and scatter correction (A). Reconstructed PET image with attenuation, but without scatter correction (B). Reconstructed PET image with both attenuation and scatter correction (C). Decomposition-based deep learning attenuation and scatter corrected PET image (D). Voxel-wise attenuation correction ratio map, dividing AC NSC PET by NAC NSC PET (E). Voxel-wise scattering correction ratio map, dividing AC SC PET by AC NSC PET (F). Voxel-wise AC SC ratio map (ADCM), dividing AC SC PET by NAC NSC PET (G). Deep learning generated ADCM (H).

We apologize for not showing the corresponding non-corrected and corrected PET together along with the attenuation map in the previous Figure 9, which we have updated to show in Figure 8.

(ii) As shown in the updated Figure 8, the lung vessels and bone structures can be seen in the PET data, and as discussed in the previous answer, the ADCM is also

correlated with intensity, so it is reasonable to reflect the activity distribution from the PET data, showing the lung vessels and bone structures in the ADCM.

Additionally, the lung vessels can be seen from the PET due to the higher sensitivity of the Siemens Quadra, while the ADCM of the Siemens Vision 450 does not show them as clearly as the Quadra (Figure 10).

And we agree that we should show the ADCM example, so we added Figure 10 into Supplementary materials (Supplementary Figure 4).

Figure 10. (Supplementary Figure 4) Example of the anatomy-dependent correction map (ADCM) and CT-based μ -map with Siemens Biograph Vision 450 ^{18}F -FDG.

(iii) We apologize for misunderstanding in the previous figure. The pseudo-CT and PET are from the same patient but not the same slice. We thank the Reviewer to point out this. Following the comment, we showed the results of pseudo μ -map based method in Figure 5. As answered above, we trained networks to synthesize pseudo μ -map rather

than pseudo-CT based on the Reviewer's comment.

(iv) As shown in Figure 11 and Table 2. According to the Pearson test, p values are larger than 0.05 on all datasets except Vision 600 (Bern)- ⁶⁸Ga-DOTA-TOC, which contains only 8 subjects. The results imply that the performance of our model was not affected by the weight distribution of the subjects, as body mass index (BMI) was generally not associated with NRMSE. Studies using the direct approach also claimed the robustness of the model in terms of correcting images obtained in patients with a high BMI ¹².

On the other side, the AI-based estimation of attenuation map or pseudo-CT is also dependent on the global anatomy and can be also influenced by BMI. There is generally no direct local mapping between PET imaging and attenuation maps. The essentials of the learning of attenuation maps from PET imaging is recognizing the anatomical structures, which can be influenced or biased by the training data due to the limited extrapolation capability. In most existing studies, the images employed during network training cannot cover all the extreme situations, such as broken skulls, or brain regions after surgery. The robustness of network predictions of these special situations requires further investigation ¹³, and training datasets representative of these patterns of abnormality are necessary for learning based algorithms.

Figure 11. Scatter plot of the relationship between body mass index and normalized root mean squared error in different scanners and tracers. (Supplementary Figure 5)

Source	Cross Scanner				Cross tracer			
	Vision 450 (SH)- FDG	Vision 600 (Bern)- FDG	UI uMI 780 (SH)- FDG	GE Discovery MI (SH)- FDG	Vision 450 (SH)- ⁶⁸ Ga-FAPI	Vision 450 (SH)- ⁶⁸ Ga-DOTA-TATE	Vision 600 (Bern)- ⁶⁸ Ga-DOTA-TOC	Vision 600 (Bern)- ¹⁸ F-PSMA
Pearson r	-0.2671	0.1836	0.03857	0.04052	0.7152	0.1845	0.8308	0.3618
P value (two-tailed)	0.0582	0.1531	0.7061	0.6830	0.0708	0.4784	0.0206	0.2478

Table 2. Pearson test relationship between body mass index (BMI) and normalized root mean squared error (NRMSE) in different scanners and tracers. BMI is not correlated with the NRMSE. (Supplementary Table S2)

4. Overall, I want to suggest the use of clear terminology that reflects what is actually shown. E.g. instead of an "anatomy dependent correction map" the term ratio map would be more intuitive. Also, the authors use terms like "low-frequency components" which are (in a strict sense) not related to the presented method. Importantly, I still cannot identify where domain knowledge is injected in this approach. Domain knowledge would be injected if e.g. a pseudo attenuation map is estimated because then the physical knowledge would be used for reconstruction. It seems to me that the proposed method in comparison rather neglects this domain knowledge resulting in the above-mentioned issues (1)

Response: Overall, we would like to thank the Reviewer to help us to think deeper about our method, especially by comparing the ADCM to the μ -map.

As for the terminology, we agree that "ratio map" is intuitively easier to understand, but we would like to have a term that reflects the idea of Decomposition here. With the decomposition, we were able to extract the anatomy dependent component from the NASC PET data. And the term of "low-frequency components" account for ADCM at lower spatial resolution ¹⁴.

The decomposition design of image components can be considered as domain transfer. After the domain transfer using decomposition, we are searching for solution in a meaningful but easier domain, which will make the solution robust and accurate. This domain transfer strategy is a typical method in many applications. For example, pathological images can be decomposed to polarization images ¹⁶, which contain abundant polarization and micro-structural information which helps to recognize lesion tissue. Similarly, spherical harmonics decomposition technique is used to infer the three-dimensional distribution from projections images ¹⁷. Likewise, PET imaging denoising was achieved by transforming a given image into a wavelet basis ¹⁸. Similar to the design of the above-mentioned studies, we decomposed the complex end-to-end generation from NASC-PET to ASC-PET into two components, anatomy-independent textures (relating to tracers and diseases) and anatomy-dependent correction. And the resulting ADCM contains abundant information for attenuation and scatter correction.

Our proposed Decomposition-based DL modified one step of the conventional 3D method, i.e., we regularized the network to estimate anatomy-dependent information derived from NASC-PET, which is a voxel-wise ratio map. The design of ratio-based method was motivated by the principle of attenuation correction. PET measurements with attenuation can be described as: $m_p = \exp(-\mathcal{R}\mu) \cdot \mathcal{R}\lambda$ ¹⁵, where m_p is non-attenuation-corrected activity sinogram, \mathcal{R} is the Radon transform, μ is the linear attenuation coefficient derived from attenuation map, λ representing spatial distributions of radioactivity concentrations. Similarly, our proposed method can be described as: $m_p = R(\text{ADCM}) \cdot \mathcal{R}\lambda$, where ADCM is the voxel-wise anatomy-dependent correction ratio map, and $R()$ is an imaginary transform similar to the

inverse of the Radon transform. In general, the ADCM integrates physical knowledge in its representations, which is more difficult for human perception but not challenging for AI. Due to the short cut in the learning, it is easier and more robust in the practice of AI-based correction.

We hope that this could clarify better the role of the domain knowledge in our method.

References

1. Marshall, H. R. *et al.* Variable lung density consideration in attenuation correction of whole-body PET/MRI. *J. Nucl. Med.* **53**, 977–984 (2012).
2. Marshall, H. R. *et al.* Description and assessment of a registration-based approach to include bones for attenuation correction of whole-body PET/MRI. *Med. Phys.* **40**, 82509 (2013).
3. Chatziioannou, A. & Dahlbom, M. Detailed investigation of transmission and emission data smoothing protocols and their effects on emission images. *IEEE Trans. Nucl. Sci.* **43**, 290–294 (1996).
4. Salimans, T. & Kingma, D. P. Weight normalization: A simple reparameterization to accelerate training of deep neural networks. *Adv. Neural Inf. Process. Syst.* **29**, (2016).
5. Sekine, T. *et al.* Evaluation of atlas-based attenuation correction for integrated PET/MR in human brain: application of a head atlas and comparison to true CT-based attenuation correction. *J. Nucl. Med.* **57**, 215–220 (2016).
6. Akbarzadeh, A., Ay, M. R., Ahmadian, A., Riahi Alam, N. & Zaidi, H. MRI-guided attenuation correction in whole-body PET/MR: assessment of the effect of bone attenuation. *Ann. Nucl. Med.* **27**, 152–162 (2013).
7. Keereman, V., Van Holen, R., Mollet, P. & Vandenberghe, S. The effect of errors in segmented attenuation maps on PET quantification. *Med. Phys.* **38**, 6010–6019 (2011).
8. Watson, C. C., Hu, J. & Zhou, C. Extension of the SSS PET scatter correction algorithm to include double scatter. in *2018 IEEE Nuclear Science Symposium and Medical Imaging Conference Proceedings (NSS/MIC)* 1–4 (IEEE, 2018).
9. Zaidi, H. & Montandon, M.-L. Scatter compensation techniques in PET. *PET Clin.* **2**, 219–234 (2007).
10. Manglos, S. H., Bassano, D. A., Duxbury, C. E. & Capone, R. B. Attenuation maps for SPECT determined using cone beam transmission computed tomography. *IEEE Trans. Nucl. Sci.* **37**, 600–608 (1990).
11. Heußner, T. *et al.* Investigation of the halo-artifact in ⁶⁸Ga-PSMA-11-PET/MRI. *PLoS One* **12**, e0183329 (2017).
12. Yang, J., Sohn, J. H., Behr, S. C., Gullberg, G. T. & Seo, Y. CT-less direct correction of

attenuation and scatter in the image space using deep learning for whole-body FDG PET: potential benefits and pitfalls. *Radiol. Artif. Intell.* **3**, (2021).

13. Gong, K., Berg, E., Cherry, S. R. & Qi, J. Machine learning in PET: from photon detection to quantitative image reconstruction. *Proc. IEEE* **108**, 51–68 (2019).
14. Quarantelli, M. *et al.* Frequency encoding for simultaneous display of multimodality images. *J. Nucl. Med.* **40**, 442–447 (1999).
15. Berker, Y. & Li, Y. Attenuation correction in emission tomography using the emission data—a review. *Med. Phys.* **43**, 807–832 (2016).
16. Zhao, Y. *et al.* Detecting giant cell tumor of bone lesions using Mueller matrix polarization microscopic imaging and multi-parameters fusion network. *IEEE Sens. J.* **20**, 7208–7215 (2020).
17. Volegov, P. L. *et al.* Three-dimensional reconstruction of neutron, gamma-ray, and x-ray sources using spherical harmonic decomposition. *J. Appl. Phys.* **122**, 175901 (2017).
18. Kang, S.-K., Yie, S.-Y. & Lee, J.-S. Noise2Noise Improved by Trainable Wavelet Coefficients for PET Denoising. *Electronics* **10**, 1529 (2021).

Reviewers' Comments:

Reviewer #2:

Remarks to the Author:

All my remaining questions in the previous round have been well addressed. Thank the authors very much.

Reviewer #3:

None